

# Unveiling community patterns and trophic niches of tropical and temperate ants using an integrative framework of field data, stable isotopes and fatty acids

Felix B. Rosumek[1,2], Nico Blüthgen[1], Adrian Brückner[1,3],
Florian Menzel[4], Gerhard Gebauer[5] and Michael Heethoff[1]

[1] Ecological Networks, Technische Universität Darmstadt, Darmstadt, Germany
[2] Department of Ecology and Zoology, Federal University of Santa Catarina, Florianópolis, Brazil
[3] Division of Biology and Biological Engineering, California Institute of Technology, Pasadena, CA, USA
[4] Institute of Organismic and Molecular Evolution, Johannes-Gutenberg Universität Mainz, Mainz, Germany
[5] BayCEER – IBG, Universität Bayreuth, Bayreuth, Germany

Corresponding authors
Felix B. Rosumek,
rosumek@hotmail.com
Michael Heethoff,
heethoff@bio.tu-darmstadt.de

## ABSTRACT

**Background:** The use and partitioning of trophic resources is a central aspect of community function. On the ground of tropical forests, dozens of ant species may be found together and ecological mechanisms should act to allow such coexistence. One hypothesis states that niche specialization is higher in the tropics, compared to temperate regions. However, trophic niches of most species are virtually unknown. Several techniques might be combined to study trophic niche, such as field observations, fatty acid analysis (FAA) and stable isotope analysis (SIA). In this work, we combine these three techniques to unveil partitioning of trophic resources in a tropical and a temperate community. We describe patterns of resource use, compare them between communities, and test correlation and complementarity of methods to unveil both community patterns and species' niches.

**Methods:** Resource use was assessed with seven kinds of bait representing natural resources available to ants. Neutral lipid fatty acid (NLFA) profiles, and $\delta^{15}N$ and $\delta^{13}C$ isotope signatures of the species were also obtained. Community patterns and comparisons were analyzed with clustering, correlations, multivariate analyses and interaction networks.

**Results:** Resource use structure was similar in both communities. Niche breadths ($H'$) and network metrics ($Q$ and $H'_2$) indicated similar levels of generalization between communities. A few species presented more specialized niches, such as *Wasmannia auropunctata* and *Lasius fuliginosus*. Stable isotope signatures and NLFA profiles also indicated high generalization, although the latter differed between communities, with temperate species having higher amounts of fat and proportions of C18:1n9. Bait use and NLFA profile similarities were correlated, as well as species' specialization indices ($d'$) for the two methods. Similarities in $\delta^{15}N$ and bait use, and in $\delta^{13}C$ and NLFA profiles, were also correlated.

**Discussion:** Our results agree with the recent view that specialization levels do not change with latitude or species richness. Partition of trophic resources alone does not explain species coexistence in these communities, and might act together with

behavioral and environmental mechanisms. Temperate species presented NLFA patterns distinct from tropical ones, which may be related to environmental factors. All methods corresponded in their characterization of species' niches to some extent, and were robust enough to detect differences even in highly generalized communities. However, their combination provides a more comprehensive picture of resource use, and it is particularly important to understand individual niches of species. FAA was applied here for the first time in ant ecology, and proved to be a valuable tool due to its combination of specificity and temporal representativeness. We propose that a framework combining field observations with chemical analysis is valuable to understand resource use in ant communities.

## INTRODUCTION

The use and partitioning of trophic resources is a central aspect of community functioning. Trophic interactions govern the flux of matter and energy in food webs, and lead to other fundamental interactions such as competition and mutualism (*Polis & Strong, 1996*; *Reitz & Trumble, 2002*). Trophic niche partitioning is one of the most important mechanisms allowing species coexistence, and may ultimately link to evolutionary processes of adaptation and character displacement (*Schluter, 2000*).

Ants (Hymenoptera: Formicidae) are among the most abundant groups of invertebrates in terrestrial ecosystems, presenting a wide range of feeding habits, nesting sites and interactions with organisms from all trophic levels. In general they are regarded as omnivorous, feeding on a combination of living prey, dead arthropods, seeds and plant exudates (*Blüthgen & Feldhaar, 2010*; *Lanan, 2014*). On the ground of tropical forests, dozens of species may coexist at the same spot, which raises the question: how ecologically different are these species? Although the role of interspecific competition in ant communities has recently been hotly debated (*Cerdá, Arnan & Retana, 2013*), the combination of high species richness with high biomass may lead to evolutionary pressure for more diversified niches. *MacArthur (1972)* suggested that specialization increases in tropical communities and, as a result, more species can coexist. However, this idea was put in question by recent studies (*Schleuning et al., 2012*; *Morris et al., 2014*; *Frank et al., 2018*). For ants, behavioral and environmental mechanisms of coexistence have been proposed (*Cerdá, Retana & Cros, 1997*; *Andersen, 2000*; *Parr & Gibb, 2012*). The use of food resources itself is surprisingly understudied, and trophic niches of most species remain poorly known. This is particularly evident in rich tropical communities (*Rosumek, 2017*), but also true for some temperate species (*Lanan, 2014*).

Field observations are the most straightforward way of gathering information, but there are trade-offs between the number of species studied (e.g., single species natural history vs. community patterns; *Medeiros & Oliveira, 2009*; *Houadria et al., 2015*),

the number of resources assessed (e.g., protein/sugar comparisons vs. all resources collected by workers; *Kaspari & Yanoviak, 2001*; *Lopes, 2007*) and the sampling intensity (e.g., seasonal studies vs. temporal "snapshots"; *Albrecht & Gotelli, 2001*; *Rosumek, 2017*). Moreover, many species present cryptic habits, and the sheer complexity of interactions makes the assessment of trophic niches a laborious task. Baiting is a method widely used in ant ecology to assess communities and infer resource use (*Bestelmeyer et al., 2000*), but it is affected by the aforementioned drawbacks.

Several techniques have been applied in ecology to deal with these issues, among them stable isotope analysis (SIA) and fatty acid analysis (FAA). Indirect techniques could be faster and reduce fieldwork effort, but also rely on several assumptions to interpret their results. Since every method has its assets and caveats, the choice depends on the nature of the questions being asked (*Birkhofer et al., 2017*). However, this also works the other way around: complementary methods can be combined to provide a detailed and integrative perspective on the community being studied.

Stable isotopes have been widely applied to address several questions in ant biology (*Feldhaar, Gebauer & Blüthgen, 2010*). Most commonly used are the relative abundance of heavy nitrogen ($\delta^{15}$N) and carbon ($\delta^{13}$C) (*Hyodo, 2015*). $\delta^{15}$N increases predictably when one organism consumes another, thus indicating whether species are at the top or bottom of the food web (*Heethoff & Scheu, 2016*). $\delta^{13}$C could be used to distinguish between main carbon sources at the bottom of the food web because $C_3$, $C_4$ and CAM plants have different signatures (*O'Leary, 1988*; *Gannes, Del Rio & Koch, 1998*). SIA provides time-representative clues about trophic position, but limited information on specific food sources or feeding behaviors. For instance, if two species feed exclusively on primary consumers, they will have similar $\delta^{15}$N, regardless of what prey items they actually consume, or whether the food is obtained through predation or scavenging. As such, stable isotope signatures are not suitable to calculate niche breadth or overlap, or to be analyzed as species-resources interaction networks.

Fatty acids obtained from the diet are mainly stored as neutral lipid fatty acids (NLFAs) in the fat body of insects. Some fatty acids can be synthesized de novo by organisms, from carbohydrates or other fatty acids. Synthesis of C16:0, C18:0 and C18:1n9 (palmitic, stearic and oleic acids) is widespread, and they are the most abundant NLFAs in insects (*Stanley-Samuelson et al., 1988*; but see *Thompson, 1973*). Ability to synthesize other NLFAs is highly variable among taxonomic groups, such as C18:2n6 (linoleic acid; *Malcicka, Visser & Ellers, 2018*). When fatty acids are reliably assigned to specific food sources, they may act as biomarkers (*Ruess & Chamberlain, 2010*). Even when such biomarkers are not identified, all fatty acids assimilated without modification (i.e., through direct trophic transfer) influence the composition of the fat reserves, including the relative amounts of de novo-synthesized NLFAs. Hence, the stored fat preserves information on ingested carbon sources, and NLFA profiles can be compared to infer niche differences (*Budge, Iverson & Koopman, 2006*). However, the application of FAA in field studies of terrestrial organisms still is limited. Most studies focused on soil detritivores, such as collembolans and nematodes (*Ruess et al., 2007*; *Haubert et al., 2009*; *Ngosong et al., 2009*). So far, FAA was not used to study trophic ecology of ants.

In this work, we combine field observations with SIA and FAA to unveil the use and partitioning of trophic resources in a tropical and a temperate epigeic ant community. Our main goal is to describe patterns for each community and test differences between communities and methods. Specifically, we aim to: (1) assess use of multiple resources, stable isotope signatures and NLFA profiles of the most abundant species in both communities; (2) compare patterns between communities using descriptive and statistical approaches; (3) test whether different methods provide convergent or complementary information on patterns of resource use.

## MATERIALS AND METHODS

### Baiting

Fieldwork in Brazil was carried out in Florianópolis (Desterro Conservation Unit, 27°31′38″S, 48°30′15″W, altitude ca. 250 m), in December 2015 and January 2016, under sampling permit SISBIO 51173-1 (ICMBio), and export permits 15BR019038/DF and 17BR025207/DF (IBAMA). The vegetation consists of a secondary Atlantic forest with at least 60 years of regeneration. High rainfall rate along the coast results in high productivity, ant species richness and a tropical aspect for the Atlantic forest even at higher latitudes such as in our work (*Silva & Brandão, 2014*). In Germany, it was carried out in Darmstadt (Prinzenberg, 49°50′14″N, 08°40′01″E, altitude ca. 250 m), in July 2015 (no permits needed there). The vegetation consists of patches of mixed forest, beech forest and orchards, which were all covered by the sample grids.

Sampling design followed similar protocols in both sites (Table 1). Seven bait types were offered as proxies for resources that are widely used by ants in general (*Kaspari, 2000*; *Blüthgen & Feldhaar, 2010*; *Lanan, 2014*; for a full description of baits, see Supplemental Document S1). Sample points were distributed in grids and separated by 10 m. In each sampling session, only a single bait was offered per point, and bait types were randomized among points. Baits were set up in transparent plastic boxes and retrieved after 90 min. This procedure was repeated in different days until all bait types had been offered at each point (twice in Brazil). The design was based on *Houadria et al. (2015)* and evaluates use of multiple resources, differing from a typical cafeteria experiment, which is designed to assess preferred resources (*Krebs, 1999*).

### Pitfall sampling

We performed a concomitant pitfall assessment to verify whether bait records represented well the epigeic community (Table 1). One vial per sample point was previously buried to avoid the digging-in effect (*Greenslade, 1973*), and replaced after collection for the next round. Pitfall and bait sampling were not performed simultaneously at the same point. Vials were buried at ground level, had six cm diameter and 150 ml volume, and contained 40 ml propylene glycol 50%.

### Fatty acid analysis

In Brazil, samples were obtained from baits and complemented by colony sampling in November 2017. We only used ants from melezitose, sucrose and seed baits, to avoid

**Table 1 Details of the sampling design applied in this study.**

| | Brazil | Germany |
|---|---|---|
| **Sampling effort** | | |
| Sampling points | 64 | 80 |
| Period of the day | Day and night | Day |
| Baits | 64 per resource per period (= 896 baits) | 80 per resource (= 560 baits) |
| Pitfall sampling | Three 10-h rounds per period (= 60 h) | Three 12-h rounds (= 36 h) |
| **Resource represented** | | |
| Larger, faster and harder prey | Living crickets (*Achaeta domesticus* Linnaeus, 1758) | |
| Smaller, slower and softer prey | Living termites (Nasutitermitinae) | Living maggots (*Lucilia sericata* Meigen, 1826) |
| Dead arthropods | Crushed crickets and maggots/mealworms (*Tenebrio molitor* Linnaeus, 1758) | |
| Bird droppings | Chicken feces from organic breeding | |
| Seeds | Seed mixture of diverse sizes and shapes, without elaiosomes | Seeds of *Chelidonium majus* (L.), with elaiosomes |
| Oligosaccharides in honeydew | Melezitose 20% | |
| Disaccharides in nectar and fruits | Sucrose 20% | |

interference of bait lipids. In Germany, they were obtained by colony sampling between July and August 2017. Samples were frozen at −18 °C directly from the field. Total lipids were extracted from the ants whole body using one ml chloroform:methanol solution, 2:1 (v/v). The solution was applied to SiOH-columns and the neutral parcel (= mono-, di- and triglycerides) eluted with four ml chloroform. Samples were analyzed with gas chromatography–mass spectrometry, following the same procedures described in *Rosumek et al. (2017)*. NLFA amounts were obtained comparing their proportions to an internal standard (C19:0 in methanol; $\rho_i$ = 220 ng/μl). Ants were subsequently dried to obtain their lean dry weight (= without lipids).

## Stable isotope analysis

Ants collected in baits and conserved in ethanol 70% were used to analyze $\delta^{15}$N and $\delta^{13}$C. C and N isotope abundances were measured in a dual element analysis mode with an elemental analyzer coupled to a continuous flow isotope ratio mass spectrometer as described in *Bidartondo et al. (2004)*. Relative abundances were calculated following the equation: $\delta_x = (R_{sample}/R_{standard} - 1) \times 1,000$ [‰], where $R$ denotes the ratio between heavy and light isotopes of samples and international standards ($N_2$ in the air and $CO_2$ in PeeDee belemnite). Gasters were removed prior to analysis to avoid interference of gut content (*Blüthgen, Gebauer & Fiedler, 2003*).

## Taxonomic identification

Ants were identified with taxonomic revisions, and comparison to identified specimens in collections and AntWeb images (*AntWeb, 2016*). Updated names were checked with

Antcat (*Bolton, 2018*). Identifications were partially confirmed by taxonomists (see Acknowledgements).

In Brazil, ants were identified to genus level with *Baccaro et al. (2015)* and to species level with: *Acanthognathus*—*Galvis & Fernández (2009)*; *Acromyrmex*—*Gonçalves (1961)*; *Cephalotes*—*De Andrade & Baroni Urbani (1999)*; *Crematogaster*—*Longino (2003)*; *Cyphomyrmex*—*Kempf (1965)* and *Snealling & Longino (1992)*; *Gnamptogenys*—*Lattke (1995)*; *Hylomyrma*—*Kempf (1973)*; *Linepithema*—*Wild (2007)*; *Octostruma*—*Longino (2013)*; *Odontomachus* and *Pachycondyla*—*Fernández (2008)*; *Pheidole*—*Wilson (2003)*; *Wasmannia*—*Longino & Fernández (2007)*. *Camponotus* and *Strumigenys* were identified solely by comparison with collections.

In Germany, ants were identified to genus and species with *Seifert (2007)*, *Seifert & Schultz (2009)* and *Radchenko & Elmes (2010)*.

All material is stored in the collections of the Ecological Networks research group, Technische Universität Darmstadt, Darmstadt, Germany and Department of Ecology and Zoology, Federal University of Santa Catarina, Florianópolis, Brazil.

## Data analysis

As a first step, we compared species' incidences in baits and pitfalls (i.e., number of sampling points where it was recorded with each method). We assumed incidence in pitfalls to represent abundance in the community, and qualitatively compared it to incidence in baits to check whether common species were underrepresented in baits. To account for different efficiencies between methods, expected incidences were indicated by a line of slope $m = I_{baits}/I_{pitfalls}$, where $I$ is the sum of all incidences for each method (*Houadria et al., 2015*).

Number of species, replicates and individuals per sample differed between methods, based on sample availability and ant size. In Brazil, we analyzed 24 species for baits, 41 for FAA and 31 for SIA. Method comparisons were performed only with 22 species considered in all three datasets. In Germany, seven species were analyzed with all methods. For a full list of recorded species and respective labels used in plots, see Table S1.

Unless noted otherwise, similarity matrices were based on unweighted Bray–Curtis dissimilarities, and Mantel tests and correlations used Spearman's coefficient (rho). Analyses were run in R 3.4.3 (*R Core Team, 2017*) and PAST 3.14 (*Hammer, Harper & Ryan, 2001*).

For all bait analyses, we used proportion of occurrence on each bait type, relative to total records for each species. Only species with at least 10 records from five or more sample points were considered. In Brazil, day and night records were considered as independent to calculate proportions. For FAA, we calculated proportions of each NLFA relative to total composition, and used average proportions for each species. All NLFAs with average proportion >0.01% were considered. For SIA, we also used species' averages and analyzed $\delta^{15}N$ and $\delta^{13}C$ separately, using Euclidean distances to build similarity matrices. A special case was *Lasius fuliginosus* in Germany, which was represented by a single colony that foraged over a large area. Bait records from different sample points were considered independent, and chemical results represent the average of different samples from that colony.
To analyze resource use, we used clustering and network analysis. UPGMA clustering was used for species, to show functional groups based on similar use of resources, and for baits, to show the structure of resource use in each community. Statistical significance of clusters was tested with SIMPROF (*Clarke, Somerfield & Gorley, 2008*) using the package "clustsig" (*Whitaker & Christman, 2015*). A Mantel test was used to compare similarity matrices of Brazil and Germany.

For network analysis, we used quantitative modularity ($Q$) (*Dormann & Strauss, 2014*) and specialization indices for species/resources ($d'$) and whole networks ($H_2'$) (*Blüthgen, Menzel & Blüthgen, 2006*), using the package "bipartite" (*Dormann, Fruend & Gruber, 2017*). Modularity shows how compartmentalized is a community, that is, if there are groups of species that strongly interact with groups of resources. In turn, $d'$ indicates whether individual species are specialized in certain resources, or resources that are used by a specialized group of species. $H_2'$ is an extension of $d'$ and shows how specialized the network is overall. $H_2' = 0$ would mean that all species used resources in the same proportions, and $H_2' = 1$ that each species has its exclusive pattern of resource use.

Specialization indices were also used to analyze species × fatty acids contingency tables. In this case, they indicate how exclusively NLFAs are distributed across species (*Brückner & Heethoff, 2017*). $H_2' = 0$ would mean that all compounds occur in the same proportion in all species, and $H_2' = 1$ that each species has its exclusive compounds. Correspondingly, relatively high $d'$ represents NLFAs that occur more exclusively in certain species, or species with more exclusive proportions of certain NLFAs. Low $d'$ means a compound that is widespread among species, or species with similarly generalized profiles. Additionally, we tested whether the two communities differed in their overall NLFA composition with PERMANOVA, using site as a fixed factor (*Anderson, 2001*). Homogeneity of multivariate dispersion was tested a priori with PERMDISP (*Anderson & Walsh, 2013*). To detect which NLFAs contributed to differences, we used SIMPER (*Clarke, 1993*). These tests were performed using package "vegan" (*Oksanen et al., 2017*).

To test whether niche breadths and NLFA profile diversity were different between communities at species level, we calculated Shannon diversity indices for each ant species as $H' = \Sigma p_i \ln p_i$, where $p_i$ is the proportion of each resource $i$ used by the species, or NLFA found in its profile, and compared them with Mann–Whitney tests.

To test whether particular NLFAs were related to use of certain resources, we performed principal component analyses (PCA) using baits × species contingency tables, replacing zeros by small values (0.000001) and using centered log-ratio transformation to deal with the constant-sum constraint (*Brückner & Heethoff, 2017*). PC axes were correlated with NLFAs using function "envfit" from package "vegan." We also compared proportions, amounts (in $\mu$g/mg; the amount of fat divided by lean dry weight) and unsaturation indices (UI; the sum of percentages of each unsaturated NLFA multiplied by its number of double bounds) between Brazil and Germany with Mann–Whitney tests. We did this for total fat and the three most abundant NLFAs (C16:0, C18:0 and C18:1n9). To test whether there was a direct relationship between total fat amount and C18:1n9, or total amount and UI, we correlated values for all individual samples of each community (166 in Brazil, 32 in Germany).

Finally, to test whether the results yielded by the three methods were correlated, we performed Mantel tests between similarity matrices of species for each method. We also correlated species' $d'$ values for baits and NLFAs, to test whether specialization levels were related.

## RESULTS

### Use of resources

Most common species were recorded in baits in proportions similar to the expected, given their frequency in the community (Fig. S1 and Table S1). A few species were underrepresented in baits (e.g., *Pachycondyla harpax*, *Myrmica scabrinodis*, *Stenamma debile*), but, in general, species with few bait records were also rare in pitfalls. Thus, we consider that a representative part of the epigeic communities was properly sampled. Despite strong variation in total number of records, the number of species recorded in each bait was similar, with exception of large prey (Table 2).

Similarities in resource use were correlated between Brazil and Germany (Fig. 1, Mantel test, rho = 0.63, $p$ = 0.03). In both communities, large prey was set apart from the other resources, being used less frequently and by fewer species. Seeds and melezitose changed positions between communities. In Germany, all ants used both sugars indiscriminately, while in Brazil several species used more sucrose (e.g., *Camponotus zenon*, *Gnamptogenys striatula*, *Pachycondyla striata*, *Odontomachus chelifer*, *Solenopsis* sp.6) and others used more melezitose (e.g., *Pheidole aper*, *Solenopsis* sp.8, *Wasmannia affinis*) (Table 2). Both modularity ($Q_{BR}$ = 0.16, $Q_{GE}$ = 0.14) and network specialization ($H'_{2BR}$ = 0.13, $H'_{2GE}$ = 0.12) were relatively low and similar between sites. Species used resources in different ways and a few were more specialized (see below), but there were no clear links between particular resources and species or groups of species.

In Brazil, *W. auropunctata* occupied a highly specialized niche, using only feces baits, which lead to the highest $d'$ values for any species and resource. *Linepithema iniquum* also showed a relatively higher specialization level due to its preference for dead arthropods and low use of sugars. *P. striata* and *O. chelifer* used more large prey, dead arthropods and sucrose. *C. zenon* grouped with them based on use of dead arthropods and sucrose, but avoided large prey. *P. aper* was the only species to have melezitose as its preferred resource. Other species showed higher redundancy and clustered together, including all *Solenopsis* and most *Pheidole* (Fig. 1; Table 2).

In Germany, only *L. fuliginosus* showed a relatively high specialization level and clustered separately, due to its almost exclusive use of animal resources (living prey and dead arthropods). Other species showed low specialization and dissimilarity (Fig. 1; Table 2).

Niche breadths were similar between communities (Mann–Whitney, $p$ = 0.44). Average Shannon index was 1.6 ± 0.4 SD in Brazil and 1.7 ± 0.1 SD in Germany (Table 2).

### Fatty acids

Temperate species contained much higher amounts of total fat than tropical ones (Fig. 2). Fatty acid compositions changed between communities (PERMANOVA, $r^2$ = 0.35,

**Table 2 Resource use of ant species in Brazil and Germany.** Values for the seven baits are given in % of the total records for each species. Only species with at least 10 records from five sample points are listed.

| Species | Large prey | Small prey | Dead arthropods | Feces | Seeds | Melezitose | Sucrose | $d'$ | Shannon index | Records |
|---|---|---|---|---|---|---|---|---|---|---|
| **Brazil** | | | | | | | | | | |
| *Camponotus zenon* | – | 14 | 36 | 7 | 7 | 7 | 29 | 0.06 | 1.6 | 14 |
| *Gnamptogenys striatula* | 2 | 23 | 21 | 19 | 11 | 6 | 17 | 0.04 | 1.8 | 47 |
| *Linepithema iniquum* | 10 | 10 | 50 | 20 | – | 10 | – | 0.16 | 1.4 | 10 |
| *Linepithema micans* | – | 6 | 31 | 6 | 19 | 19 | 19 | 0.03 | 1.7 | 16 |
| *Nylanderia* sp.1 | 7 | 10 | 25 | 6 | 9 | 23 | 21 | 0.03 | 1.8 | 267 |
| *Odontomachus chelifer* | 26 | 5 | 19 | 5 | 5 | 10 | 31 | 0.12 | 1.7 | 42 |
| *Pachycondyla striata* | 14 | 3 | 42 | – | 1 | 6 | 34 | 0.16 | 1.3 | 88 |
| *Pheidole aper* | 4 | – | 15 | 19 | 7 | 37 | 19 | 0.08 | 1.6 | 27 |
| *Pheidole lucretii* | 4 | 4 | 26 | 8 | 14 | 20 | 24 | 0.02 | 1.8 | 50 |
| *Pheidole nesiota* | 4 | 9 | 20 | 4 | 16 | 25 | 21 | 0.02 | 1.8 | 89 |
| *Pheidole sarcina* | 4 | 8 | 16 | 14 | 20 | 18 | 22 | 0.01 | 1.9 | 51 |
| *Pheidole sigillata* | 4 | 10 | 24 | 10 | 16 | 13 | 22 | 0.00 | 1.8 | 91 |
| *Pheidole* sp.1 | 3 | 13 | 19 | 10 | 18 | 17 | 21 | 0.01 | 1.9 | 101 |
| *Pheidole* sp.2 | 6 | 11 | 14 | 14 | 21 | 20 | 15 | 0.01 | 1.9 | 322 |
| *Pheidole* sp.4 | 5 | 6 | 21 | 19 | 14 | 14 | 21 | 0.01 | 1.9 | 78 |
| *Pheidole* sp.7 | 6 | 6 | 6 | 6 | 41 | 18 | 18 | 0.08 | 1.6 | 17 |
| *Solenopsis* sp.1 | 4 | 14 | 18 | 13 | 26 | 12 | 13 | 0.01 | 1.8 | 141 |
| *Solenopsis* sp.2 | 2 | 14 | 24 | 7 | 28 | 13 | 13 | 0.03 | 1.8 | 180 |
| *Solenopsis* sp.3* | – | 4 | 16 | 8 | 32 | 20 | 20 | 0.05 | 1.6 | 25 |
| *Solenopsis* sp.4 | 1 | 5 | 21 | 10 | 31 | 14 | 18 | 0.03 | 1.7 | 96 |
| *Solenopsis* sp.6 | 2 | 10 | 29 | 7 | 17 | 10 | 26 | 0.02 | 1.7 | 42 |
| *Solenopsis* sp.8 | 7 | 11 | 29 | 7 | 25 | 14 | 7 | 0.03 | 1.8 | 28 |
| *Wasmannia affinis* | – | 25 | 10 | 10 | 30 | 20 | 5 | 0.09 | 1.6 | 20 |
| *Wasmannia auropunctata** | – | – | – | 100 | – | – | – | 0.62 | 0 | 19 |
| $d'$ | 0.17 | 0.09 | 0.09 | 0.24 | 0.14 | 0.07 | 0.09 | $H'_2 = 0.13$ | | |
| Total richness[†] | 26 | 31 | 33 | 32 | 32 | 34 | 34 | | | |
| Total records[†] | 107 | 203 | 422 | 215 | 344 | 327 | 366 | | | |
| **Germany** | | | | | | | | | | |
| *Formica fusca* | 4 | 5 | 21 | 2 | 12 | 26 | 30 | 0.06 | 1.6 | 57 |
| *Lasius fuliginosus* | 20 | 27 | 33 | 13 | – | – | 7 | 0.31 | 1.5 | 15 |
| *Lasius niger* | 7 | 14 | 19 | 11 | 9 | 19 | 20 | 0.01 | 1.9 | 118 |
| *Lasius platythorax* | – | – | 17 | 17 | 4 | 30 | 30 | 0.11 | 1.5 | 23 |
| *Myrmica rubra* | 3 | 13 | 15 | 13 | 3 | 25 | 30 | 0.03 | 1.7 | 40 |
| *Myrmica ruginodis* | – | 10 | 27 | 14 | 6 | 18 | 24 | 0.03 | 1.7 | 49 |
| *Temnothorax nylanderi* | – | 4 | 21 | 14 | 18 | 21 | 22 | 0.06 | 1.7 | 165 |
| $d'$ | 0.29 | 0.14 | 0.02 | 0.05 | 0.13 | 0.11 | 0.05 | $H'_2 = 0.12$ | | |
| Total richness[†] | 4 | 8 | 8 | 8 | 7 | 9 | 11 | | | |
| Total records[†] | 14 | 42 | 99 | 56 | 54 | 102 | 116 | | | |

**Notes:**
 * Species not considered in comparisons between methods.
 † Including species with less than 10 records.
 –, Species not recorded in this bait.

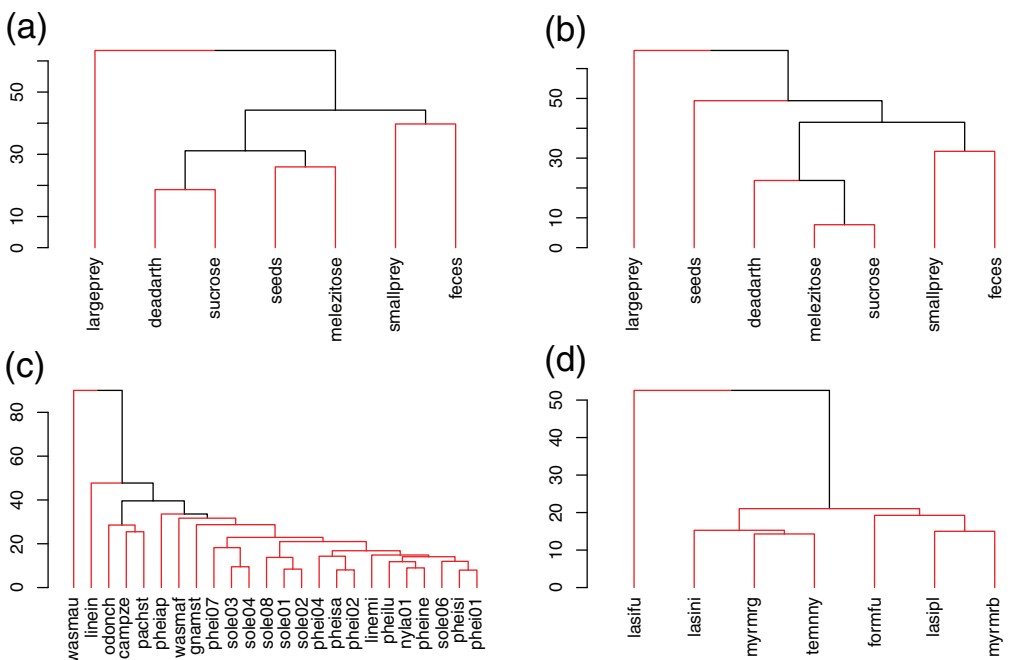

**Figure 1 UPGMA clustering of resources and species in Brazil (A, C) and Germany (B, D), based on Bray–Curtis dissimilarities.** Red lines link elements from the same statistically significant cluster (SIMPROF, $p < 0.05$).

$p < 0.01$). Multivariate dispersion was heterogeneous, being higher in Brazil than in Germany (PERMDISP, $F = 11.32$, $p < 0.01$). This does not change the previous result because, in this case, PERMANOVA becomes overly conservative (*Anderson & Walsh, 2013*).

The main reason for this difference was the predominant role of C18:1n9 in temperate species (SIMPER, dissimilarity contribution = 47%, $p < 0.01$, Figs. 2 and 3; Table 3). In Brazil, composition was more balanced, which led to higher proportions of C18:0 (contribution = 21%, $p < 0.01$), although amounts were similar. C16:0 was proportionally the most abundant NLFA in Brazil and the difference from Germany was marginally significant (contribution = 20%, $p = 0.06$), although amounts again were higher in temperate species. A few other NLFAs had statistically significant, but very small contributions to the difference (Table S2).

Fatty acid compositions were generalized overall, but more homogeneous in Germany because of the predominance of C18:1n9 ($H'_{2\,\text{BR}} = 0.09$, $H'_{2\,\text{GE}} = 0.03$). Accordingly, NLFA profile diversity was higher in tropical species (average Shannon index = $1.5 \pm 0.2$ SD) than temperate ones ($0.9 \pm 0.2$ SD) (Mann–Whitney, $p < 0.01$) (Fig. 3; Table 3).

In samples from Germany, there was no correlation between total amount of fat and percentage of C18:1n9 (rho = 0.19, $p = 0.30$) or unsaturation index (rho = 0.24, $p = 0.89$). In samples from Brazil, there was weak negative correlation between total fat and both C18:1n9 (rho = $-0.16$, $p = 0.04$) and unsaturation index (rho = $-0.22$, $p > 0.01$).

In Brazil, several fatty acids were related to resource use (Fig. 4; see Table S3 for PCA eigenvalues and full Envfit results). Species with higher C18:1n9 also used more dead

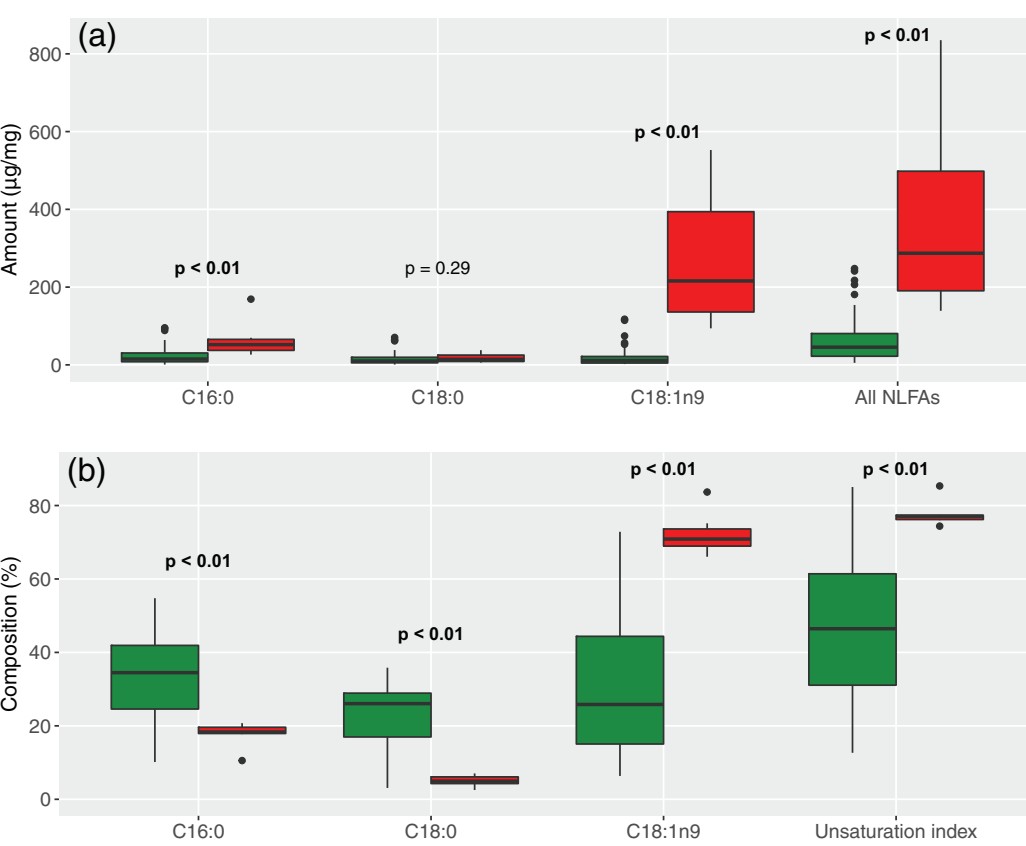

**Figure 2 Comparison of the three most abundant NLFAs, total amounts and unsaturation indices between tropical and temperate species.** (a) Amounts; (b) percentages. Green = tropical species; red = temperate species. Significant differences are in bold (Mann–Whitney test).

arthropods (Envfit, $r^2$ of the NLFA with PC axes = 0.32, $p$ = 0.02), while C18:2unk1 (an unidentified NLFA) was related to use of sucrose and large prey ($r^2$ = 0.31, $p$ = 0.03). C14:0 (mystric acid) tended to be higher in species that used more seeds, feces and small prey ($r^2$ = 0.39, $p$ = 0.01). Notice that the first two Principal Components explained only 60% of the variance and linear regressions were not strong. In Germany, most variation was along the sugar-protein axis. C18:0 and C17:0 (margaric acid) were strongly correlated with PC axes ($r^2$ = 0.84, $p$ = 0.05 and $r^2$ = 0.78, $p$ = 0.04, respectively). Both were higher in species that used more sugars, and C17:0 also was related to use of feces, although its relative abundance was very low in all species (<0.5%, Table 3).

### Stable isotopes

In Brazil, *W. auropunctata* presented distinctive signatures for both isotopes. It was the species with highest $\delta^{15}N$, while most species ranged from 5.8 to 8.2, and six showed conspicuously lower signatures. Besides *W. auropunctata*, $\delta^{13}C$ varied less, ranging from −24.1 to −27 (Fig. 5; Table 4).

In Germany, $\delta^{15}N$ was lower overall, ranging from 3.6 (*Lasius niger*) to −1.1 (*Lasius fuliginosus*). Species varied little in $\delta^{13}C$ (from −25.4 to −26.3), with values within the range of Brazilian species (Fig. 5; Table 4).

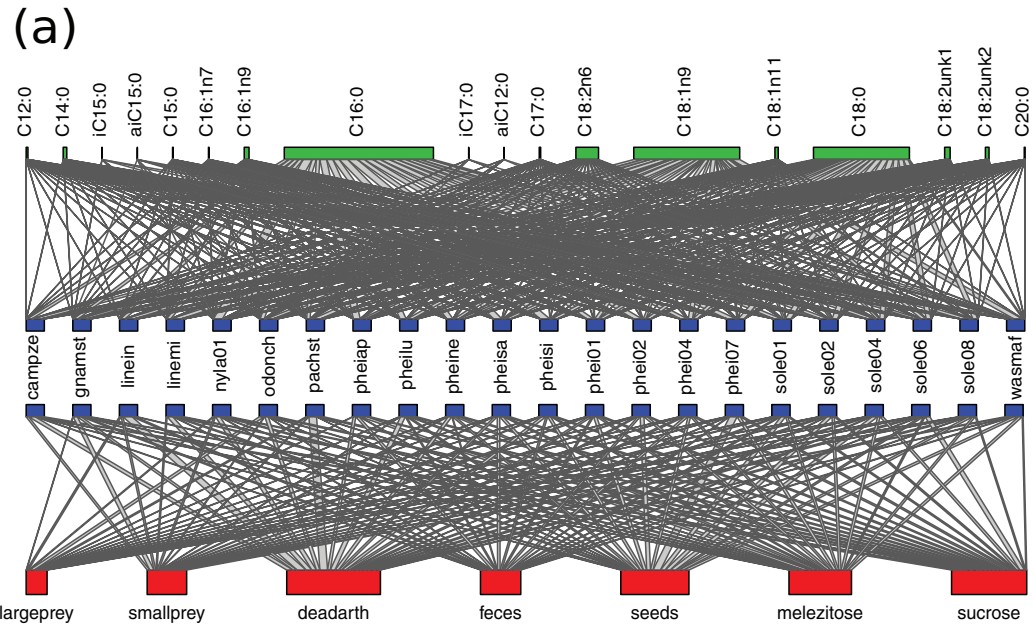

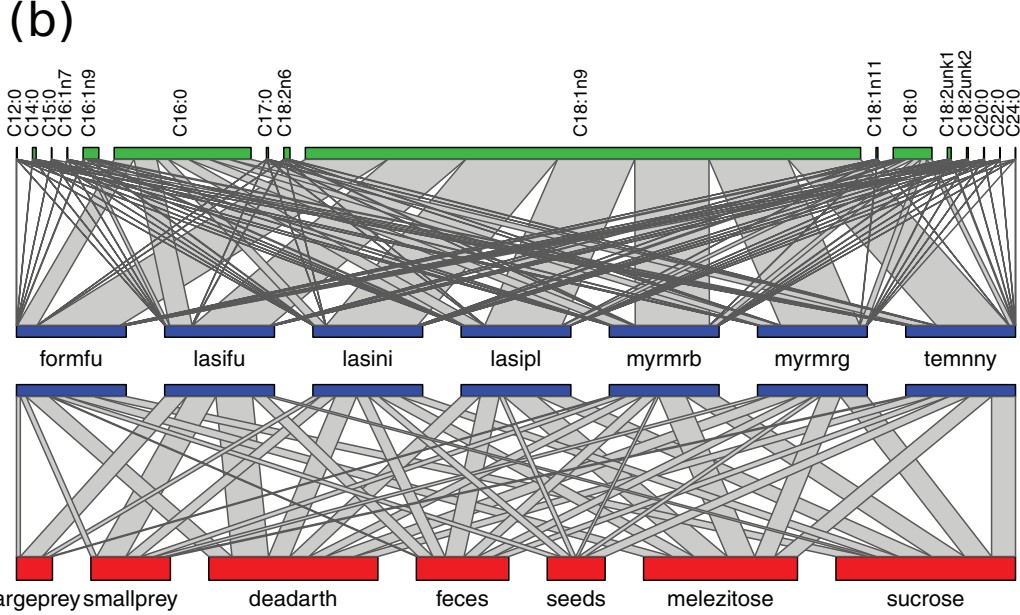

**Figure 3 Networks of resource use and fatty acid composition in Brazil (A) and Germany (B).** Species labels are standardized to represent 100% of resource use/NLFA composition. The width of connecting lines represents proportion of bait use/NLFA abundance for each species. Bait/NLFA labels show the sum of proportions for the whole community. Only species analyzed with all three methods are included.

Isotope signatures were correlated for tropical species (rho = 0.43, $p$ = 0.02). For temperate species, the correlation lacked statistical significance (rho = −0.46, $p$ = 0.3), but their inclusion slightly strengthened the correlation for all species together (rho = 0.47, $p$ < 0.01).

**Table 3 Fatty acid profiles of ant species in Brazil and Germany.** Average individual NLFA abundances are given in % of the total composition.

| Species | C12:0 | C14:0 | C15:0 | aiC15:0 | iC15:0 | C16:0 | C16:1n7 | C16:1n9 | C17:0 | aiC17:0 | iC17:0 | C18:0 | C18:1n9 | C18:1n11 | C18:2n6 | C18:2un1 | C18:2un2 | C20:0 | C22:0 | C24:0 | d' | Shannon | Amount (µg/mg) | UI | Samples |
|---|---|---|---|---|---|---|---|---|---|---|---|---|---|---|---|---|---|---|---|---|---|---|---|---|---|
| **Brazil** | | | | | | | | | | | | | | | | | | | | | | | | | |
| Acanthognathus brevicornis* | 0.3 | 1.0 | 0.4 | – | 0.3 | 28 | 0.3 | 0.3 | 0.5 | 0.04 | 0.1 | 24 | 24 | 2.0 | 11 | 3.6 | 2.8 | 0.8 | – | – | 0.01 | 1.8 | 60 | 61 | 2 |
| Acanthognathus ocellatus* | 0.4 | 0.9 | 0.3 | 0.01 | 0.04 | 38 | 0.1 | 0.4 | 0.4 | – | – | 30 | 21 | 1.1 | 5.4 | 1.4 | 1.0 | 0.5 | – | – | 0.01 | 1.5 | 65 | 38 | 2 |
| Acromyrmex aspersus* | 0.3 | 1.3 | 0.1 | – | – | 31 | 0.1 | 4.3 | 0.1 | – | – | 21 | 29 | 1.2 | 5.7 | 2.4 | 1.8 | 0.8 | – | – | 0.01 | 1.7 | 13 | 55 | 2 |
| Acromyrmex laticeps* | 0.2 | 0.2 | 0.1 | – | – | 10 | – | 0.2 | 0.5 | – | – | 16 | 37 | 1.1 | 23 | 6.0 | 5.1 | 0.5 | – | – | 0.09 | 1.7 | 8 | 106 | 1 |
| Acromyrmex lundii* | 0.2 | 0.4 | 0.0 | – | – | 13 | – | 0.4 | 0.2 | – | – | 21 | 44 | 1.8 | 11 | 4.2 | 3.7 | 0.4 | – | – | 0.04 | 1.6 | 6 | 84 | 1 |
| Acromyrmex subterraneus* | 0.1 | 0.2 | 0.1 | 0.01 | 0.03 | 10 | 0.2 | 0.5 | 0.4 | 0.03 | 0.01 | 18 | 44 | 1.2 | 17 | 4.0 | 3.6 | 0.5 | – | – | 0.06 | 1.6 | 30 | 95 | 2 |
| Azteca sp.2* | 0.5 | 0.3 | 0.0 | – | – | 13 | 0.1 | 0.4 | 0.1 | – | – | 10 | 73 | 0.8 | 1.0 | 0.7 | 0.5 | 0.2 | – | – | 0.10 | 1.0 | 62 | 78 | 3 |
| Camponotus lespesii* | 0.3 | 0.4 | 0.2 | – | – | 30 | 0.2 | 1.4 | 0.1 | – | – | 18 | 48 | 0.6 | 0.6 | 0.3 | 0.2 | 0.2 | – | – | 0.03 | 1.3 | 11 | 52 | 2 |
| Camponotus zenon | 0.6 | 1.0 | 0.2 | – | – | 28 | 0.2 | 2.8 | 0.3 | – | – | 15 | 50 | 0.8 | 0.8 | 0.2 | 0.1 | 0.1 | – | – | 0.03 | 1.3 | 5 | 56 | 2 |
| Cephalotes pallidicephalus* | 0.1 | 0.8 | 0.0 | 0.1 | 0.1 | 25 | – | 6.0 | 0.1 | – | – | 3 | 60 | 4.4 | 0.5 | – | – | 0.4 | – | – | 0.11 | 1.2 | 95 | 71 | 2 |
| Cephalotes pusillus* | 0.7 | 1.0 | 0.2 | – | – | 17 | 3.1 | 2.5 | 0.3 | – | – | 12 | 61 | 1.9 | 0.4 | – | – | 0.8 | – | – | 0.09 | 1.3 | 36 | 69 | 1 |
| Crematogaster nigropilosa* | 0.2 | 1.0 | 0.0 | – | – | 27 | 0.3 | 0.5 | 0.2 | – | – | 17 | 48 | 0.6 | 3.1 | 1.1 | 0.6 | 0.4 | – | – | 0.02 | 1.4 | 154 | 59 | 6 |
| Cyphomyrmex rimosus* | 0.5 | 1.6 | 0.2 | – | – | 38 | 0.1 | 2.7 | 0.3 | 0.04 | – | 34 | 15 | 1.0 | 3.7 | 1.0 | 0.8 | 0.5 | – | – | 0.02 | 1.5 | 86 | 30 | 4 |
| Gnamptogenys striatula | 0.2 | 1.0 | 0.3 | 0.01 | 0.3 | 27 | 0.5 | 1.1 | 0.6 | 0.1 | 0.2 | 18 | 39 | 2.4 | 6.0 | 1.9 | 1.4 | 0.7 | – | – | 0.01 | 1.7 | 56 | 61 | 6 |
| Hylomyrma reitteri* | 0.7 | 1.1 | – | – | – | 55 | – | – | 0.4 | – | – | 29 | 9 | 0.6 | 3.0 | 1.1 | 0.6 | – | – | – | 0.05 | 1.2 | 22 | 19 | 1 |
| Linepithema iniquum | 0.1 | 0.8 | 0.1 | – | – | 26 | 0.4 | 4.2 | 0.2 | – | – | 15 | 47 | 0.9 | 2.6 | 1.1 | 0.7 | 0.4 | – | – | 0.02 | 1.5 | 248 | 62 | 3 |
| Linepithema micans | 0.4 | 0.7 | 0.1 | – | – | 24 | 0.1 | 0.2 | 0.2 | – | – | 16 | 56 | 0.5 | 1.4 | 0.3 | 0.2 | 0.2 | – | – | 0.04 | 1.2 | 94 | 61 | 4 |
| Nylanderia sp.1 | 0.4 | 0.7 | 0.2 | 0.01 | – | 49 | 0.1 | 2.1 | 0.2 | – | – | 28 | 15 | 0.7 | 3.1 | 0.4 | 0.3 | 0.1 | – | – | 0.03 | 1.3 | 31 | 25 | 11 |
| Octostruma petiolata* | 0.5 | 1.4 | 0.3 | 0.1 | 0.2 | 42 | 0.3 | 1.4 | 0.8 | 0.1 | 0.1 | 36 | 12 | 1.2 | 2.0 | 0.6 | 0.4 | 0.5 | – | – | 0.03 | 1.4 | 85 | 21 | 4 |
| Odontomachus chelifer | 0.2 | 0.7 | 0.2 | 0.1 | 0.1 | 23 | 0.3 | 1.9 | 0.6 | 0.1 | 0.1 | 16 | 38 | 1.7 | 9.4 | 4.2 | 2.5 | 0.8 | – | – | 0.02 | 1.8 | 17 | 74 | 10 |
| Pachycondyla harpax* | 0.3 | 0.5 | 0.1 | 0.1 | 0.1 | 19 | 0.1 | 0.3 | 0.6 | – | – | 22 | 40 | 0.8 | 9.0 | 3.1 | 2.9 | 0.3 | – | – | 0.02 | 1.6 | 12 | 72 | 1 |
| Pachycondyla striata | 0.1 | 0.5 | 0.1 | 0.02 | 0.1 | 19 | 0.1 | 1.2 | 0.4 | 0.03 | 0.1 | 13 | 46 | 1.1 | 13 | 3.7 | 2.0 | 0.4 | – | – | 0.04 | 1.6 | 47 | 85 | 16 |
| Pheidole aper | 0.6 | 0.8 | 0.1 | – | – | 38 | 0.1 | 0.3 | 0.3 | – | – | 25 | 23 | 0.3 | 9.1 | 1.5 | 1.3 | – | – | – | 0.01 | 1.5 | 27 | 47 | 9 |
| Pheidole lucretii | 0.3 | 0.7 | 0.2 | – | – | 34 | 0.1 | 0.4 | 0.4 | – | – | 21 | 32 | 1.2 | 5.7 | 1.9 | 1.5 | 0.2 | – | – | <0.01 | 1.5 | 59 | 52 | 12 |
| Pheidole nesiota | 0.4 | 0.7 | 0.1 | – | – | 35 | 0.1 | 0.3 | 0.3 | – | – | 27 | 25 | 0.4 | 6.4 | 2.1 | 1.6 | 0.1 | – | – | <0.01 | 1.5 | 58 | 46 | 6 |
| Pheidole risii* | 0.6 | 0.7 | 0.2 | – | – | 41 | 0.1 | 0.3 | 0.3 | – | – | 30 | 19 | 0.5 | 5.1 | 1.0 | 0.8 | 0.1 | – | – | 0.01 | 1.4 | 38 | 34 | 1 |
| Pheidole sarcina | 0.5 | 0.8 | 0.1 | – | – | 47 | 0.1 | 0.2 | 0.3 | – | – | 32 | 13 | 0.5 | 4.1 | 0.8 | 0.6 | 0.2 | – | – | 0.03 | 1.3 | 20 | 24 | 1 |
| Pheidole sigillata | 0.7 | 1.1 | 0.2 | – | – | 53 | 0.1 | 0.3 | 0.7 | – | – | 34 | 6 | 0.9 | 1.7 | 0.5 | 0.3 | 0.1 | – | – | 0.06 | 1.2 | 46 | 13 | 1 |
| Pheidole sp.1 | 0.5 | 0.6 | 0.1 | – | – | 37 | 0.1 | 0.3 | 0.5 | – | – | 28 | 23 | 0.5 | 6.3 | 1.8 | 1.1 | 0.1 | – | – | <0.01 | 1.5 | 18 | 42 | 1 |

(Continued)

| Species | C12:0 | C14:0 | C15:0 | aiC15:0 | iC15:0 | C16:0 | C16:1n7 | C16:1n9 | C17:0 | aiC17:0 | iC17:0 | C18:0 | C18:1n9 | C18:1n11 | C18:2n6 | C18:2unk1 | C18:2unk2 | C20:0 | C22:0 | C24:0 | d' | Shannon | Amount | UI (µg/mg) | Samples |
|---|---|---|---|---|---|---|---|---|---|---|---|---|---|---|---|---|---|---|---|---|---|---|---|---|---|
| *Pheidole* sp.2 | 0.5 | 0.9 | 0.2 | – | – | 47 | 0.1 | 0.3 | 0.3 | – | – | 28 | 14 | 0.6 | 6.5 | 0.9 | 0.6 | 0.1 | – | – | 0.02 | 1.4 | 18 | 31 | 4 |
| *Pheidole* sp.4 | 0.7 | 0.8 | 0.3 | <0.01 | 0.01 | 40 | 0.1 | 0.9 | 0.2 | – | – | 26 | 23 | 0.8 | 3.8 | 2.0 | 0.5 | 0.1 | – | – | <0.01 | 1.5 | 24 | 38 | 7 |
| *Pheidole* sp.5* | 1.9 | 1.0 | – | – | – | 27 | – | – | 1.0 | – | – | 26 | 30 | 1.6 | 6.6 | 3.2 | 2.3 | – | – | – | 0.01 | 1.7 | 34 | 55 | 2 |
| *Pheidole* sp.6* | 0.4 | 0.7 | 0.1 | – | – | 34 | – | 0.3 | 0.3 | – | – | 28 | 26 | 0.5 | 7.4 | 1.5 | 0.8 | 0.05 | – | – | <0.01 | 1.5 | 43 | 46 | 4 |
| *Pheidole* sp.7 | 0.4 | 0.8 | 0.1 | – | – | 52 | 0.1 | 0.1 | 0.2 | – | – | 34 | 7 | 0.4 | 4.1 | 0.8 | 0.3 | 0.1 | – | – | 0.06 | 1.2 | 181 | 18 | 4 |
| *Solenopsis* sp.1 | 0.6 | 1.8 | 0.3 | 0.03 | – | 44 | 0.1 | 0.4 | 0.6 | – | – | 32 | 11 | 0.7 | 7.7 | 0.4 | 0.3 | 0.2 | – | – | 0.03 | 1.4 | 217 | 29 | 5 |
| *Solenopsis* sp.2 | 0.7 | 1.6 | 0.4 | 0.03 | – | 43 | 0.04 | 0.4 | 0.5 | – | – | 31 | 11 | 0.6 | 8.6 | 0.8 | 0.7 | 0.2 | – | – | 0.03 | 1.5 | 206 | 33 | 5 |
| *Solenopsis* sp.4 | 0.7 | 0.6 | 0.1 | <0.01 | 0.05 | 45 | 0.1 | 0.6 | 0.2 | 0.01 | 0.03 | 26 | 18 | 0.5 | 6.7 | 0.9 | 0.7 | 0.1 | – | – | 0.01 | 1.4 | 79 | 35 | 4 |
| *Solenopsis* sp.6 | 0.4 | 0.6 | 0.2 | – | – | 36 | 0.1 | 0.5 | 0.3 | – | – | 27 | 27 | 0.4 | 4.6 | 1.2 | 0.9 | 0.3 | – | – | <0.01 | 1.5 | 41 | 42 | 3 |
| *Solenopsis* sp.8 | 0.5 | 1.0 | 0.1 | – | – | 53 | – | 0.3 | 0.4 | – | – | 27 | 11 | 0.3 | 5.3 | 1.1 | 0.7 | 0.1 | – | – | 0.04 | 1.3 | 75 | 26 | 1 |
| *Strumigenys denticulata** | 0.5 | 1.5 | 0.3 | – | – | 38 | – | 0.2 | 0.6 | – | – | 32 | 20 | 1.2 | 3.0 | 1.3 | 0.9 | 1.0 | – | – | 0.01 | 1.5 | 81 | 31 | 5 |
| *Wasmannia affinis* | 0.2 | 1.6 | 0.1 | <0.01 | 0.02 | 21 | 0.2 | 7.6 | 0.2 | 0.01 | 0.01 | 7 | 47 | 2.2 | 7.9 | 2.0 | 1.5 | 0.4 | – | – | 0.06 | 1.6 | 241 | 80 | 5 |
| *d'* | <0.01 | <0.01 | <0.01 | <0.01 | <0.01 | 0.07 | <0.01 | 0.15 | <0.01 | <0.01 | 0.01 | 0.05 | 0.13 | 0.04 | 0.09 | 0.07 | 0.08 | <0.01 | – | – | $H'_2 = 0.03$ | | | | |
| **Germany** | | | | | | | | | | | | | | | | | | | | | | | | | |
| *Formica fusca* | 0.03 | 0.1 | 0.02 | – | – | 18 | 0.1 | 0.5 | 0.1 | – | – | 4.9 | 75 | 0.01 | 0.4 | 0.2 | 0.1 | 0.1 | 0.1 | 0.01 | <0.01 | 0.8 | 287 | 77 | 5 |
| *Lasius fuliginosus* | 0.1 | 0.4 | 0.05 | – | – | 21 | 0.3 | 3.4 | 0.03 | – | – | 2.5 | 71 | 0.1 | 0.6 | 0.7 | 0.1 | 0.02 | – | – | <0.01 | 0.9 | 230 | 77 | 2 |
| *Lasius niger* | 0.05 | 0.1 | 0.01 | – | – | 11 | 0.04 | 1.1 | 0.04 | – | – | 4.0 | 84 | 0.3 | 0.1 | 0.03 | 0.02 | 0.02 | 0.01 | – | 0.02 | 0.6 | 660 | 85 | 5 |
| *Lasius platythorax* | 0.1 | 0.2 | 0.03 | – | – | 18 | 0.1 | 2.1 | 0.4 | – | – | 7.0 | 70 | 0.6 | 0.3 | 0.2 | 0.1 | 0.1 | 0.03 | 0.02 | <0.01 | 1.0 | 336 | 74 | 5 |
| *Myrmica rubra* | 0.2 | 0.6 | – | – | – | 19 | 0.1 | 1.8 | 0.2 | – | – | 6.6 | 68 | 0.2 | 1.8 | 1.0 | 0.6 | 0.2 | 0.1 | 0.03 | <0.01 | 1.1 | 139 | 76 | 5 |
| *Myrmica ruginodis* | 0.03 | 0.4 | – | – | – | 18 | – | 1.6 | 0.5 | – | – | 5.6 | 72 | 0.2 | 0.8 | 0.5 | 0.3 | 0.2 | 0.1 | 0.01 | <0.01 | 0.9 | 151 | 77 | 5 |
| *Temnothorax nylanderi* | 0.2 | 1.5 | <0.01 | – | – | 20 | 0.1 | 3.8 | 0.4 | – | – | 4.5 | 66 | 0.2 | 1.7 | 0.9 | 0.3 | 0.1 | 0.03 | 0.01 | <0.01 | 1.1 | 835 | 76 | 5 |
| *d'* | <0.01 | <0.01 | <0.01 | – | – | 0.01 | <0.01 | 0.04 | <0.01 | – | – | 0.01 | 0.01 | <0.01 | <0.01 | <0.01 | <0.01 | <0.01 | <0.01 | <0.01 | $H'_2 = 0.03$ | | | | |

**Notes:**
* Species not considered in comparisons between methods.
–, NLFA not detected in this species; UI, unsaturation index.

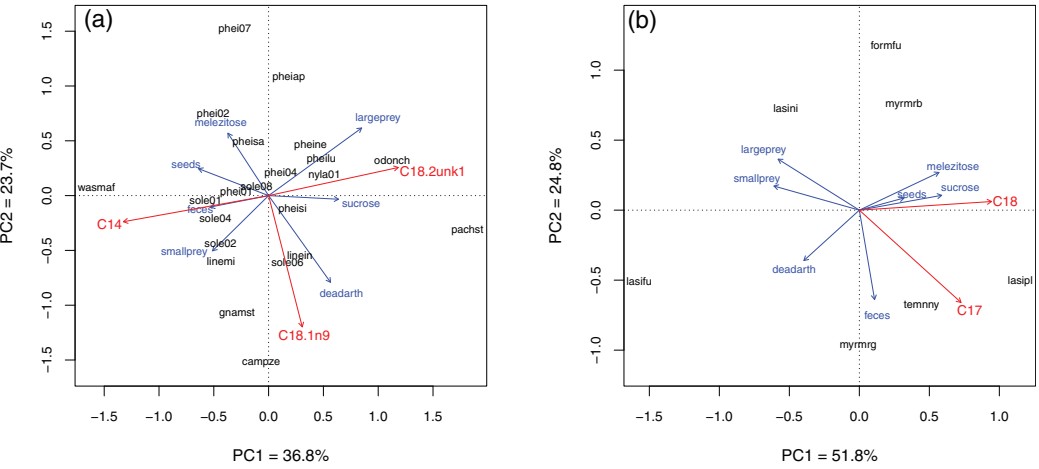

**Figure 4 Principal component analysis of resource use in Brazil (A) and Germany (B).** Blue setae = bait direction vectors. Red setae = NLFAs correlated to the two main PC axes (Envfit; only statistically significant relationships are plotted). Setae sizes indicate relative strength of the relationships, but are scaled independently for each plot. Only species analyzed with all three methods are included.

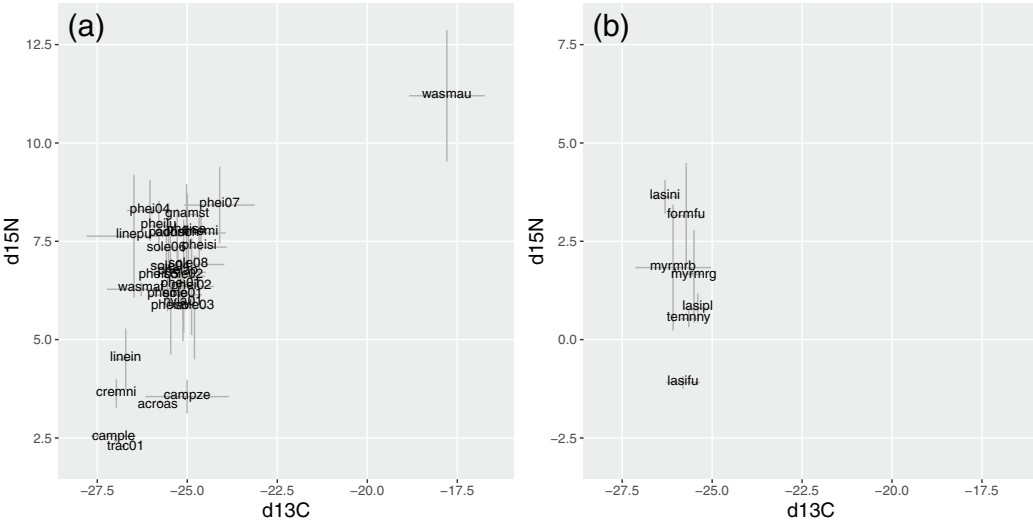

**Figure 5 Stable isotope signatures of species in Brazil (A) and Germany (B).** Gray bars display standard deviations for species averages. $\delta^{15}N$ and $\delta^{13}C$ are displayed in ‰, following the equation described in the methods.

## Dataset comparisons

In Brazil, similarities in bait use and NLFA profiles were correlated (Table 5). While $\delta^{15}N$ similarities were correlated with similarities in bait use, but not with NLFAs, the opposite was found for $\delta^{13}C$. That is, similar use of resources among species was reflected in similar body fat composition, and both were related to their long-term trophic position, albeit in different ways. In Germany, no such correlations were found between datasets (although it was marginally significant for NLFAs and $\delta^{15}N$).

**Table 4 Stable isotope signatures of ant species in Brazil and Germany.** Average $\delta^{15}N$ and $\delta^{13}C$ are given in ‰, following the equation described in the methods.

| Species | $\delta^{15}N$ | $\delta^{13}C$ | Samples |
|---|---|---|---|
| **Brazil** | | | |
| *Acromyrmex aspersus** | 3.42 | −25.76 | 1 |
| *Camponotus lespesii** | 2.44 | −26.99 | 3 |
| *Camponotus zenon* | 3.50 | −25.06 | 4 |
| *Crematogaster nigropilosa** | 3.63 | −26.97 | 2 |
| *Gnamptogenys striatula* | 8.18 | −25.00 | 5 |
| *Linepithema iniquum* | 4.50 | −26.71 | 5 |
| *Linepithema micans* | 7.71 | −24.62 | 4 |
| *Linepithema pulex** | 7.63 | −26.48 | 4 |
| *Nylanderia* sp.1 | 5.94 | −25.12 | 5 |
| *Odontomachus chelifer* | 7.69 | −25.28 | 5 |
| *Pachycondyla striata* | 7.69 | −25.52 | 5 |
| *Pheidole aper* | 6.71 | −25.26 | 5 |
| *Pheidole avia** | 5.84 | −25.46 | 3 |
| *Pheidole lucretii* | 7.89 | −25.79 | 3 |
| *Pheidole nesiota* | 6.13 | −25.55 | 5 |
| *Pheidole sarcina* | 7.77 | −25.02 | 4 |
| *Pheidole sigillata* | 7.35 | −24.67 | 5 |
| *Pheidole* sp.1 | 6.40 | −25.18 | 5 |
| *Pheidole* sp.2 | 6.35 | −24.88 | 5 |
| *Pheidole* sp.4 | 8.23 | −25.98 | 5 |
| *Pheidole* sp.5* | 6.63 | −25.79 | 1 |
| *Pheidole* sp.7 | 8.42 | −24.10 | 5 |
| *Solenopsis* sp.1 | 6.14 | −25.12 | 5 |
| *Solenopsis* sp.2 | 6.61 | −25.10 | 5 |
| *Solenopsis* sp.3* | 5.82 | −24.80 | 3 |
| *Solenopsis* sp.4 | 6.81 | −25.47 | 5 |
| *Solenopsis* sp.6 | 7.30 | −25.58 | 5 |
| *Solenopsis* sp.8 | 6.91 | −24.97 | 3 |
| *Trachymyrmex* sp.1* | 2.29 | −26.77 | 1 |
| *Wasmannia affinis* | 6.28 | −26.28 | 4 |
| *Wasmannia auropunctata** | 11.20 | −17.79 | 4 |
| **Germany** | | | |
| *Formica fusca* | 3.15 | −25.72 | 5 |
| *Lasius fuliginosus* | −1.09 | −25.81 | 5 |
| *Lasius niger* | 3.63 | −26.31 | 5 |
| *Lasius platythorax* | 0.80 | −25.40 | 5 |
| *Myrmica rubra* | 1.78 | −26.03 | 5 |
| *Myrmica ruginodis* | 1.66 | −25.56 | 5 |
| *Temnothorax nylanderi* | 0.53 | −25.65 | 5 |

**Notes:**
* Species not considered in comparisons between methods.

**Table 5 Correlations between methods in Brazil and Germany.** Results are for Mantel tests using Spearman's rho, based on similarities matrices (Bray–Curtis for baits and NLFAs, Euclidean distances for isotopes). Asterisks indicate significant correlations.

| Method | Baits | | NLFAs | |
|---|---|---|---|---|
| | rho | $p$ | rho | $p$ |
| **Brazil** | | | | |
| NLFAs | 0.43 | <0.01* | | |
| $\delta^{13}C$ | 0.23 | 0.07 | 0.23 | 0.02* |
| $\delta^{15}N$ | 0.25 | 0.04* | 0.14 | 0.08 |
| **Germany** | | | | |
| NLFAs | −0.23 | 0.77 | | |
| $\delta^{13}C$ | −0.24 | 0.76 | 0.29 | 0.19 |
| $\delta^{15}N$ | 0.37 | 0.12 | 0.46 | 0.06 |

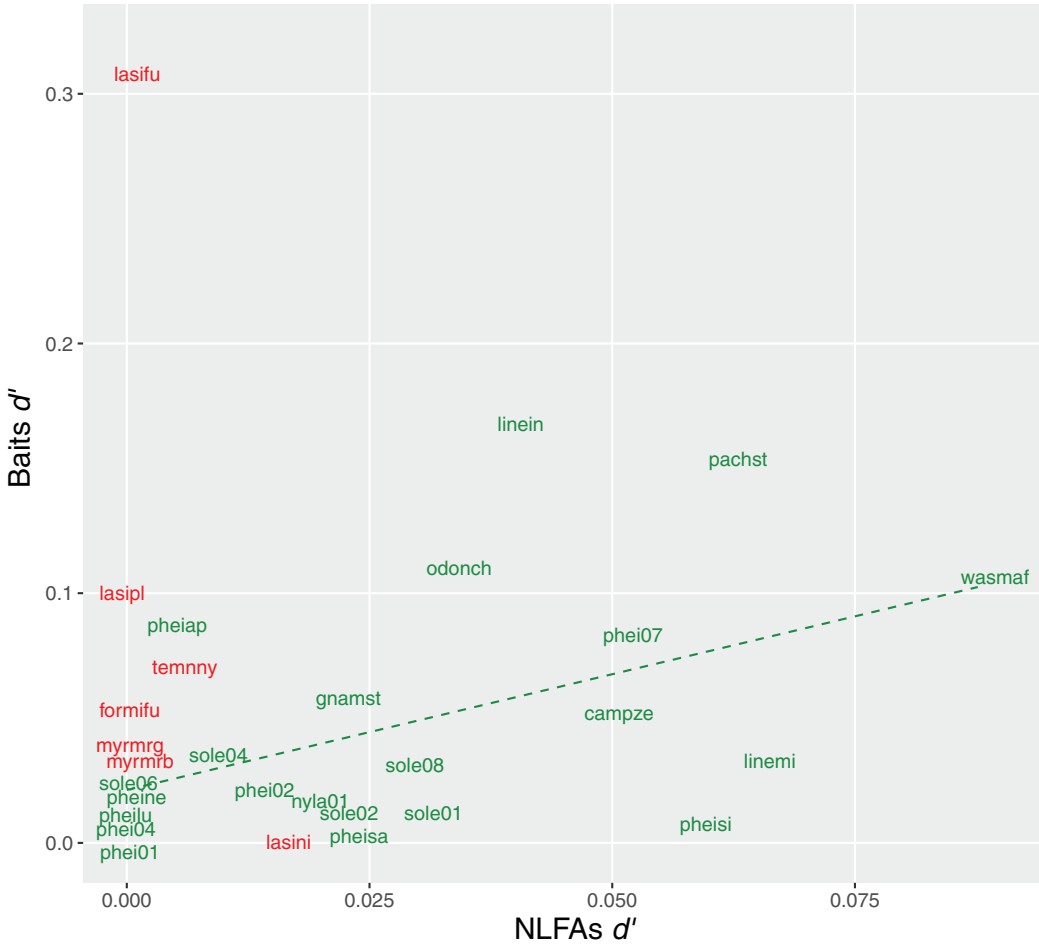

**Figure 6 Relationship between specialization indices ($d'$) for bait use and NLFAs.** Green = tropical species, the dashed line indicates significant correlation; red = temperate species, no correlation observed. Only species analyzed with all three methods are included.

Exclusiveness ($d'$) of bait choices and NLFAs profiles were also correlated in Brazil (rho = 0.51, $p$ = 0.02), suggesting that specialization in resources was reflected in more specific compositions of fatty acids. No correlation was observed in Germany (rho = −0.07, $p$ = 0.86) (Fig. 6).

## DISCUSSION

Our main findings in this work are: (1) patterns of resource use are similar in both communities, although the role of oligosaccharides is distinct; (2) both communities are similarly generalized in resource use, regardless of species richness; (3) temperate ants present higher amounts of fat and more homogeneous NLFA compositions; (4) composition and specialization in resource use and NLFAs are correlated, and are also related to species' trophic position; (5) some species show specialized behaviors that can be better understood by method complementarity.

The hypothesis proposed by *MacArthur (1972)* suggested that specialization is higher in tropical communities because the environmental stability allows species to adapt to more specialized niches without increasing extinction risk, thus allowing more species to coexist. However, this idea was put in question by recent studies, where the latitude-richness-specialization link was not confirmed, or an inverse trend was found (*Schleuning et al., 2012*; *Morris et al., 2014*; *Frank et al., 2018*). Our work is not an explicit test of this hypothesis, but several results agree with the view that specialization does not necessarily increase with higher richness toward the tropics: despite the different number of species, network metrics of resource use and niche breadths were similarly generalized in both communities; fatty acid compositions were also highly generalized, although in this case in different level, possibly due to other factors (see discussion on fatty acids below); cluster analysis of resource use showed similar patterns between communities and both species clusters and stable isotopes indicated strong overlap inside each community.

The bait protocol we applied is efficient to assess niches of generalists, and specialized species were seldom recorded. Nevertheless, these generalists represent the majority of the communities (as highlighted by our pitfall data), and one might expect more diversified niches to allow coexistence, but that was not the case. The differences we observed might still play a role in coexistence of some species, particularly when they share other traits, such as *O. chelifer* and *Pachycondyla striata*. Both are large, solitary foraging Ponerinae species, very common on the ground of the Atlantic forest, but *O. chelifer* is more predatory and *Pachycondyla striata* is more scavenging (*Rosumek, 2017*). Coexistence is result of a complex interplay of habitat structure, interspecific interactions and species traits and no single factor governs ant community organization (*Cerdá, Arnan & Retana, 2013*). Trophic niche alone does not explain coexistence of the common species in these two communities, but likely is one of the many factors structuring them.

### Use of resources

Resource use in Brazil was discussed in detail in *Rosumek (2017)*, as well as the literature review on trophic niche of our identified tropical species. Large prey was the less used

resource, because size and mobility of the prey limits which species are able to overcome them. Small prey and feces were also relatively less used, the first because also is relatively challenging to acquire, and the second probably due to smaller nutritional value. On the other hand, the other resources are nutritive and relatively easy to gather, particularly dead arthropods, which was by far the most used resource. Considering the similarity in resource use patterns, most general remarks in that work apply to Germany as well. The two main differences we found are discussed below.

The role of insect-synthesized oligosaccharides seems to be distinct between temperate and tropical communities. In Brazil, the aversion to melezitose showed by some species could represent a physiological constraint, since tolerance to oligosaccharides differs among ant species (*Rosumek, 2017*). For ants without physiological constraints, melezitose use might be opportunistic and does not necessarily mean that they interact with sap-sucking insects. However, honeydew is the only reliable source of oligosaccharides in nature, so the few species that preferred this sugar may engage in such interactions (particularly *Pheidole aper*). In Germany, on the other hand, all species used both sugars similarly. The two *Myrmica*, *Lasius* and *Formica fusca* are known to interact with sap-sucking insects, and *Temnothorax nylanderi* uses honeydew opportunistically when droplets fall on the ground (*Seifert, 2007*).

Seeds were other resource used differently, but this probably is consequence of our methodological choice of seeds with elaiosomes in Germany. Elaiosomes are thought to mimic animal prey and attract predators and scavengers (*Hughes, Westoby & Jurado, 1994*), not only granivores. Effectively, elaiosomes of *Chelidonium majus* are attractive to a wide range of ants (*Reifenrath, Becker & Poethke, 2012*). However, seeds were more extensively used in Brazil. A higher diversity of shape and sizes of seeds was offered there, which allowed more ants to use them.

## Fatty acids

Fatty acid compositions were generalized, but differed between communities. In Germany, C18:1n9 plays a prominent role, making up for more than 70% of the NLFAs stored by ants. The amounts of fat also differed remarkably: in average, temperate ants stored over five times more fat. Similarly high amounts of total fat and percentages of C18:1n9 were observed in laboratory colonies of *F. fusca* and *M. rubra* (*Rosumek et al., 2017*), which suggest that it might be a general trend for temperate species. In Brazil, NLFA abundance at community level was more balanced between C16:0, C18:0 and C18:1n9. Both amounts and proportions of C18:1n9 were variable among species.

Organisms can actively change their fatty acid composition in response to environmental factors and physiological needs (*Stanley-Samuelson et al., 1988*). Temperature and balance between saturated and unsaturated NLFAs are important, because the fat body should present a certain fluidity that allows enzymes to access stored nutrients (*Ruess & Chamberlain, 2010*). C18:1n9 seems to be the only unsaturated fatty acid that ants are able to synthesize by themselves in large amounts (*Rosumek et al., 2017*). However, there was no positive correlation between amount of fat and C18:1n9 or unsaturation index of samples, which would be expected if C18:1n9 synthesis was a direct

mechanism of individuals to balance saturation:unsaturation ratios (the weak negative correlation in Brazil also does not fit this hypothesis).

Therefore, we suggest that differences in C18:1n9 percentages and total amounts could be consequence of two distinct environmental factors. Under lower temperatures, higher proportion of unsaturated fatty acids is needed to maintain lipid fluidity (*Jagdale & Gordon, 1997*). Thus, temperate species might be adapted to synthesize and store more C18:1n9 to withstand the cold seasons. If this hypothesis were correct, the ants would maintain this high proportion throughout the year, since we collected in summer. In turn, high amount of fat could be a direct consequence of the marked seasonality in temperate regions. These species might be adapted to quickly acquire and accumulate energy reserves during the short warm season, while there is less pressure for this in regions where resources are available throughout the year.

We observed relationships between certain NLFAs and resources, although overall they were not strong and not necessarily result of direct trophic transfer. C18:1n9 was related to use of dead arthropods in Brazil. This NLFA is considered a "necromone," a chemical clue for recognition of corpses by ants and other insects (*Sun & Zhou, 2013*), so it presumably increases in dead arthropods. However, only polyunsaturated fatty acids can be degraded to form C18:1n9 during decomposition, and C18:1n9 itself turns into C18:0 (*Dent, Forbes & Stuart, 2004*). Thus, for high C18:1n9 to be a direct result of scavenging, prey items should previously possess high levels of unsaturation. This might be an indirect effect as well: scavenger ants might be better at tracking and retrieving food items that are naturally rich in C18:1n9. No correlation was found in Germany, which could also be related to the special role of this NLFA in temperate species: its predominance due to environmental factors may override its dietary signal.

C18:2n6 occurs independently of diet only in very small amounts, and it is a potential biomarker (*Rosumek et al., 2017*). The differences we observed among species are direct result of diet. Its occurrence was more widespread in Brazil, but we observed no clear correlation with specific resources. C18:2n6 is found in elaiosomes, seeds and other arthropods in different amounts (*Thompson, 1973*; *Hughes, Westoby & Jurado, 1994*). Since it can come from different sources, C18:2n6 cannot be straightforwardly used as a biomarker for specific diets, but depends on a deeper analysis of the resources actually available in the habitat.

The biological significance of the correlations of NLFAs and resources in Germany is difficult to grasp. C18:0 does not appear to be preferably synthesized from carbohydrates, compared to C16:0 and C18:1n9 (*Rosumek et al., 2017*). Adding to the fact that such correlation was not found in Brazil, this might not represent a physiological link between sugar consumption and C18:0 synthesis. With low number of species in Germany, even strong correlations might be result of species-specific factors other than diet. The same might be said for C17:0, a fatty acid that occurs in very low amounts in several vegetable oils (*Beare-Rogers, Dieffenbacher & Holm, 2001*).

Interestingly, we did not observe any 18:3n3 or 18:3n6 ($\alpha$- and $\gamma$-linolenic acids). Ants are not able to synthesize them, and they are assimilated through direct trophic

transfer (*Rosumek et al., 2017*). In the studied communities, these fatty acids seem to be completely absent from food sources used by ants. This is an unexpected result, since their occurrence is well documented in elaiosomes and a wide range of insect groups that might serve as prey (*Thompson, 1973*; *Hughes, Westoby & Jurado, 1994*).

The use of fatty acids as biomarkers to track food sources is one of the greatest potentials of this method. However, it might be more suitable to detritivore systems, where the biomarkers are distinctive membrane phospholipids from microorganisms that decompose specific resources, and that end up stored in the fat reserves of the consumers (*Ruess & Chamberlain, 2010*). The NLFA profiles we observed are generalized, and the most relevant fatty acids could represent distinct sources and/or be synthesized de novo in large amounts. The biomarker approach might not be suitable at community level for ground ants, contrary to NLFA profiles (see Method Comparison below). However, it still might be useful to unveil species-specific interactions, or in contexts with less potential sources that can be better tracked (e.g., leaf-litter or subterranean species).

## Stable isotopes

Trophic shift (i.e., the degree of change in isotopic ratios from one trophic level to another) varies among taxonomic groups and according to other physiological factors (*McCutchan et al., 2003*). "Typical" values of ca. 3‰ for $\delta^{15}$N and 1‰ for $\delta^{13}$C were experimentally observed in one ant species (*Feldhaar, Gebauer & Blüthgen, 2010*). Establishing discrete trophic levels is unrealistic in most food webs, particularly for omnivores such as ants (*Polis & Strong, 1996*), but species within the range of one trophic shift are more likely to use resources in a similar way. $\delta^{15}$N ranges of ca. 9‰ were observed for ant communities in other tropical forests, representing three trophic shifts (*Davidson et al., 2003*; *Bihn, Gebauer & Brandl, 2010*). This is similar to our range of 8.9‰ but, discounting *W. auropunctata*, the remaining range of 6.1‰ is more similar to what was observed in an Australian forest (7.1‰; *Blüthgen, Gebauer & Fiedler, 2003*). In Germany, only *Lasius fuliginosus* presented a distinct signature. In both communities, most species fell within the range of one trophic shift.

$\delta^{13}$C showed smaller, but meaningful, variations that were correlated to $\delta^{15}$N. $\delta^{13}$C is less applied to infer trophic levels, as it is more sensitive to sample preservation method and diet composition (*Tillberg et al., 2006*; *Heethoff & Scheu, 2016*). An average change of 0.61‰ was observed in samples stored in ethanol by *Tillberg et al. (2006)*. However, we observed correlations (including with NLFAs—see below) despite this eventual change, and it would not affect the similarity among species and between communities. Primary consumers using distinct plant sources may present differences of up to 20‰, and this will influence the signature of secondary consumers (*O'Leary, 1988*; *Gannes, Del Rio & Koch, 1998*). However, in our case, only *W. auropunctata* presented such distinct value.

Again, both isotopes suggest that the core of these communities is composed by generalists that broadly use the same resources. Since we did not establish baselines, lower values in Germany do not necessarily mean lower trophic levels in this community. Isotope signatures for the same species are highly variable among sites in Europe

(*Fiedler et al., 2007*), and this variation can be the result of either different isotope baselines or actual changes in species' ecological roles.

Low $\delta^{15}$N suggest that a species obtain most of their nitrogen from basal trophic levels, mainly plant sources (*Blüthgen, Gebauer & Fiedler, 2003*; *Davidson et al., 2003*). This fits the six species with lowest $\delta^{15}$N in Brazil. Two were fungus-growing ants (*Acromyrmex aspersus, Trachymyrmex* sp.1), which use mostly plant material to grow its fungus. The others were species that forage frequently on vegetation, besides the ground (*Camponotus lespesii, Camponotus zenon, Crematogaster nigropilosa, Linepithema iniquum*). Arboreal species that heavily rely on nectar or honeydew usually present low $\delta^{15}$N, which may be the case for these species. *Linepithema* represents well this trend: the two mainly ground-nesting species, *Linepithema micans* and *Linepithema pulex*, presented higher signatures than the plant-nesting *Linepithema iniquum* (*Wild, 2007*).

## Community patterns and method comparison

The correlations we observed between methods are interesting from both the methodological and the biological perspective. From a methodological viewpoint, for terrestrial animals, this is the first time an empirical relationship is shown between patterns of resource use and composition of stored fat in natural conditions, and that both relate to their long-term trophic position. Although differences between species were small, these relationships were robust enough to be detected by different methods. From a biological viewpoint, it highlights several physiological mechanisms involved in such relationships. We will discuss in the following some of these mechanisms, as well as caveats that are often cited for these methods. They probably still influence our results and correlations, but did not completely override the patterns.

A commonly cited caveat for using baits is that ants could be attracted to the most limited resources, instead of the ones they use more often. Evidence for this comes mainly from nitrogen-deprived arboreal ants (*Kaspari & Yanoviak, 2001*), and some cases are discussed below (see method complementarity). However, this effect might be less pronounced in epigeic species, and our results suggest that there is convergence between bait attendance, and medium- and long-term use of resources.

Diet may significantly change NLFA composition in a few weeks (*Rosumek et al., 2017*) and persist for a similar time (*Haubert, Pollierer & Scheu, 2011*). Therefore, the "snapshot" of resource use we observed with baits should represent at least the seasonal preferences of the species. A seasonal study on NLFA compositions can bring valuable information on resource use changes, or if they are stable throughout the year.

Adult ants are thought to feed mostly on liquid foods, due to the morphology of the proventriculus, which prevents solid particles to pass from the crop to the midgut (*Eisner & Happ, 1962*). Larvae are able to process solids and possess a more diversified suit of enzymes, and are sometimes called the "digestive caste" of the colony (*Hölldobler & Wilson, 1990*; *Erthal, Peres Silva & Ian Samuels, 2007*). Trophallaxis is an important mechanism of food sharing between workers and larvae. Our results suggest that the trophic signal of NLFAs is not lost in this processes, and that might be true even for solid

items such as arthropods or seeds. However, the similarities could be as well the result of direct digestion and assimilation of liquid sources (sugars, hemolymph).

We also found correlations with stable isotopes. They were weaker than between baits and NLFAs, and different for each isotope. For $\delta^{15}$N, it shows that patterns of resource use are more correlated with trophic level. Protein amino acids must be obtained from diet or synthesized from other nitrogenated compounds, so the signal relative to nitrogen sources should be more preserved. This also fits to the idea that $\delta^{15}$N reflects larval diet, because it is in this stage that ants grow and build most of their biomass (*Blüthgen & Feldhaar, 2010*). On the other hand, it makes sense that the signal relative to carbon sources is related to NLFA composition. We should note that we removed the gaster of the ants used in SIA, so we observed only the signal of carbon incorporated in the other body parts. This is related to dietary carbon, but a stronger signal could be expected if the fat body is included.

The low source-specificity of stable isotope signatures might also lead to relatively weak correlations. *Pachycondyla striata* and *O. chelifer* had the same $\delta^{15}$N despite their different preferences. Other species that appear to be mostly scavengers had similar or higher $\delta^{15}$N than those two "predators," such as *Linepithema micans*, *Pheidole sarcina*, *Pheidole lucretii* and *Pheidole* sp.4.

In Germany, no correlation was observed between methods. This is probably a consequence of the low number of species available in the community. The relationships found in Brazil might be valid for other communities, although ecological context and physiology might change their significance or strength.

## Species niches and method complementarity

Niche differences were correlated at community level, but the use of different techniques allows better understanding of species' niches. Method complementarity is particularly important if one is interested in the functional role of individual species, not only in overall patterns. Some cases are described below.

In Brazil, *W. auropunctata* was distinct from the remaining community, both in resource use and isotopic signature (unfortunately, no NLFA samples were obtained for this species). Strong preference for feces is a novel behavior for this species, known to invade and dominate disturbed habitats, but less dominant inside forests (*Rosumek, 2017*). Its isotopic signature confirms that they have a highly differentiated diet, and could be direct result of a feces-rich diet. In herbivorous mammals, feces are usually enriched in $\delta^{15}$N relative to diet (*Sponheimer et al., 2003*; *Hwang, Millar & Longstaffe, 2007*). The proposed mechanism of $^{15}$N enrichment along trophic levels states that this happens due to preferential excretion of $^{14}$N, and it is assumed that most nitrogen is excreted in the urine, which is depleted in $^{15}$N (*Peterson & Fry, 1987*; *Gannes, Del Rio & Koch, 1998*; but see *Sponheimer et al., 2003*). However, $^{15}$N-enriched feces were also observed in uricotelic organisms, such as birds and locusts (*Webb, Hedges & Simpson, 1998*; *Bird et al., 2008*). Thus, high $\delta^{15}$N is consistent with a diet based on $^{15}$N-enriched feces from other consumers. The relationship with the $\delta^{13}$C signature is less clear, but it also suggests high specialization.

This behavior might be a local adaptation, but also could indicate that *W. auropunctata* shifts to less disputed resources inside native forests (although the exact resources used may be context-dependent). In *Davidson (2005)*, *Wasmannia* species (including *W. auropunctata*) presented relatively high $\delta^{15}$N and were considered highly carnivorous. However, our result shows that high $\delta^{15}$N should not be taken from granted to represent high-level consumers. It might be a solid generalization for communities, but other trophic pathways may lead to such signatures. Due to their lack of specificity, isotope signatures should be combined with field observations to provide reliable information at species level. As another example, the second highest $\delta^{15}$N in our work was observed in *Pheidole* sp.7, a species that used mainly seeds and was seldom recorded in animal (or feces) baits.

Another example where results seem to be contradictory is *L. fuliginosus*, which showed strong preference for animal baits, but low $\delta^{15}$N. In this case, the natural history of the species is well known, and it strongly interacts with aphids, particularly the giant oak aphid *Stomaphis quercus* (*Seifert, 2007*). This suggests that this aphid's honeydew is not enriched in $^{15}$N and has a composition similar to the plants on which they feed. The honeydew supply should be abundant, since ants basically ignored sugar baits, but also relatively poor in nitrogen, which makes *L. fuliginosus* use animal sources whenever possible. A similar pattern may apply to *Linepithema iniquum* in Brazil, which also combined low $\delta^{15}$N with preference for animal baits. This species is also known to use extra-floral nectaries and honeydew (*Rosumek, 2017*), but does not have such strong and specific interactions as its temperate counterpart.

## CONCLUSIONS

In this work, we investigated two communities with three distinct methods, and provided information on community patterns of resource use and species' trophic niches. Our results agree with the view that ant communities are mostly composed by generalist species that share similar resources, and suggest that such patterns do not differ between tropical and temperate communities. Although high richness may lead to more specialists in the tropics, the generalist core of the community should be maintained by a combination of several factors.

Overall, we observed that the three methods corresponded in their characterization of the communities, but their combination provided a more comprehensive picture of resource use. However, the time and costs demanded should limit the broad application of this framework, and some techniques are more suitable to answer particular questions. We gave special focus on FAA in this work because it was the first time this method was applied to study ant ecology in the field. Considering that NLFA profiles provide a more time-representative snapshot than baits, and are more specific than stable isotopes, we suggest FAA as a powerful tool to study trophic niche relationships in species-rich ant communities. It allows the researcher to obtain quantitative data related to diet with relatively short fieldwork time, or from systems where direct observation is limited, and then use it to infer niche breadths, similarities and overlap. However, their use as biomarkers has yet to be developed, and seems to be limited for epigeic ant communities.

Combining NLFA compositions with field observations is strongly recommended if the researcher is interested in source-specificity. Finally, stable isotopes (particularly $\delta^{15}N$) might be added as a long-term representation of trophic position, which can corroborate or complement other results.

## ACKNOWLEDGEMENTS

We thank Cristian Klunk, Frederico Rottgers Marcineiro, Larissa Zanette da Silva, Lukas Kauling and Felix Schilcher for assistance in fieldwork and sample sorting; Anna Ruppenthal for assistance in fieldwork and FAA of German ants; Christine Tiroch for technical assistance in EA-IRMS; and Rodrigo Machado Feitosa, Alexandre Casadei Ferreira, Thiago Sanches Ranzani da Silva and Francisco Hita Garcia for taxonomic assistance. We also thank the two anonymous reviewers for their valuable comments and suggestions.

### Funding

Felix B. Rosumek and Adrian Brückner were supported by PhD scholarships from the Brazilian National Council of Technological and Scientific Development (CNPq), and the German National Academic Foundation (Studienstiftung des deutschen Volkes), respectively. This study was funded by CNPq (290075/2014-9), the German Research Foundation (DFG; HE 4593/5-1) and the Open Access Publishing Fund of DFG/Technische Universität Darmstadt. The funders had no role in study design, data collection and analysis, decision to publish or preparation of the manuscript.

### Grant Disclosures

The following grant information was disclosed by the authors:
Brazilian National Council of Technological and Scientific Development (CNPq): 290075/2014-9.
German National Academic Foundation (Studienstiftung des deutschen Volkes.
German Research Foundation (DFG): HE 4593/5-1.
Open Access Publishing Fund of DFG/Technische Universität Darmstadt.

### Competing Interests

The authors declare that they have no competing interests.

### Author Contributions

- Felix B. Rosumek conceived and designed the experiments, performed the experiments, analyzed the data, contributed reagents/materials/analysis tools, prepared figures and/or tables, authored or reviewed drafts of the paper, approved the final draft.
- Nico Blüthgen conceived and designed the experiments, contributed reagents/materials/analysis tools, authored or reviewed drafts of the paper, approved the final draft.
- Adrian Brückner conceived and designed the experiments, analyzed the data, authored or reviewed drafts of the paper, approved the final draft.

- Florian Menzel conceived and designed the experiments, authored or reviewed drafts of the paper, contributed reagents/materials/analysis tools, approved the final draft.
- Gerhard Gebauer performed the experiments, contributed reagents/materials/analysis tools, approved the final draft.
- Michael Heethoff conceived and designed the experiments, contributed reagents/materials/analysis tools, authored or reviewed drafts of the paper, approved the final draft.

## Field Study Permissions

The following information was supplied relating to field study approvals (i.e., approving body and any reference numbers):

Fieldwork in Brazil was carried out under sampling permit SISBIO 51173-1 (ICMBio), and export permits 15BR019038/DF and 17BR025207/DF (IBAMA). No permits were needed in Germany.

## Data Availability

All raw data used for the analysis of this article are provided in Tables 2–4, and Table S1.

## Supplemental Information

Supplemental information for this article can be found online at http://dx.doi.org/10.7717/peerj.5467#supplemental-information.

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
