# Peer review of "Unveiling community patterns and trophic niches of tropical and temperate ants using an integrative framework of field data, stable isotopes and fatty acids"

_PeerJ, doi:10.7717/peerj.5467_

## Round 0.1 · original submission · Major Revisions

Although both reviewers were enthusiastic about your study, they indicated a number of important issues that need to be addressed before the paper can be accepted.

Reviewer 1 ·

Basic reporting

The manuscript is well written with sufficient literature references. I particularly liked the interpretation of the Wasmannia auropunctata result (high 15N, feces feeder) in the context of Davidson (2005).

Line 72: How ‘ecologically’ different are these species?
Line 74: Replace ‘evolutive’ with ‘evolutionary’
Line 213: How compartmentalized a community is.

Experimental design

Line 127-131: Please provide more ecological detail about the study sites.

Line 138: How does this method uniquely estimate ‘multidimensional nutritional niches’? Meaning unclear.
Line 140: It is frustrating as a reader to be referred to another paper for description of the methods used in the present study (this is a recurring theme throughout the methods). Please at least include these important details in a supplementary appendix here.
Line 143-145: Please provide more information about the pitfall sampling regime. How long were pitfalls left in the ground, and at what densities were they placed? And, how was this data used to ‘verify whether bait records represented well the epigeic community?’ Table 1 doesn’t provide enough information to assess methods or the sampling effort. Each bait type needs to be explained.
Line 149: why only this subset of baits?

Line 108-116 & 148-153: After these descriptions of the NLFA approach, it is still unclear exactly what the data mean and how they enable one to infer dietary niche breadth, and which aspects limit their utility in comparisons across species and sites. Perhaps it could help to integrate some of the detail currently developed in the Discussion (lines 403-473) in the context of hypotheses in the Introduction?
Additionally, given the newness of this technique to the field, it is not sufficient to refer the reader to the Rosumek study for the details in the Methods.
Line 151: redefine NLFA here.

Line 156: Aren’t ants stored in EtOH of limited utility for SIA analyses? Was any attempt made to verify the effects of the storage method on the results of isotopic analyses?
Line 160 (also 494-499): To what extent the SIA values presented here comparable across communities? Aren’t such values usually baselined relative to potential food resources (and plant material at the 1st trophic level) available in each community? Moreover, the underlying logic of using heavy isotope signatures to make ecological inference should be expanded upon (i.e. move some of the ideas from the Discussion [Lines 475-509] to the Introduction in the context of a priori hypotheses).

Line 179: where is the reference collection stored? And, how were the Solenopsis sp. morphospecies determined?
Line 182: Unclear why bait data were combined with pitfall data for analyses.
Line 184-186: Meaning unclear. How were expected proportions weighted? How would this influence the results?

Line 195 & Table 2: Table 2 needs information on the sampling units. Were 14 Camponotus zenon workers observed across a certain number of baits (how many baits were provided of each resource per site?) or from one colony on a single bait? How are these numbers comparable? Are these percents? If so, percents of what? How is a Shannon index generated for each species? Define d’… There is too little information here to clearly interpret these data.

Line 196-197: Please provide more details about how this was done and why.
Line 197-198: Is there some way to provide information about the amount of FAA variation that was observed within species? And how were means calculated across all the NLFAs?

Line 201: Is there some concrete evidence that this was a single colony?

Line 232-234: This methodology is unclear: ‘pi is the proportion of resource use ‘i’ by each species, or NLFA found its profile’… Please elaborate

Line 240: when was it described that total fat content was measured? How many individuals per species?

Line 240-241: What is the ecological relevance of an unsaturation index [i.e. other than ‘the sum of percentages of each unsaturated NLFA multipled by its number of double bonds, Line 240-241)]?

Line 243: what is the meaning of C16:0, C18:0, C18:ln9? For the uninitiated, it is not possible to understand what these numbers mean, why they were chosen for further analysis, and what they say about ant nutritional ecology.

Validity of the findings

General speaking, the methods were sufficiently difficult to parse, as to make the results extremely difficult to interpret (see comments above).

Line 287: The amount of background information provided about this method to this point, makes it exceedingly difficult to interpret this statement.

Figure 1: It is very difficult to make sense of these trees. Are the units on the Y axis comparable between Brazil and Germany? Perhaps it would help if the colors of the tips for c and d were somehow color coded by genus or some other a priori ecological/functional trait framework?

Additional comments

This paper tackles important questions of diet breadth and community structure, comparing a German and Brazilian site. It is laudable that this study pairs 3 methods (field observation, fatty acid analysis, stable isotopes) acknowledging that each has limitations, and that together they can provide a picture of an ant’s trophic niche. Additionally, the idea that fatty acid analyses may also help corroborate stable isotope analyses is exciting.

However, this paper was also frustrating to read. Specifically, it lacked a clear conceptual framework (e.g. the methods are not clearly framed within hypotheses in the Introduction), had ambiguous methodological details (e.g. sampling design & effort) and lacked well-developed figure/table captions. Additionally, the meaning of the NLFA’s and their ecological significance within and between sites was unclear, even after reading the extended section in the Discussion. This lack of detail made it very difficult to interpret the results.

In general, perhaps the Introduction can spell out more clearly the differences between the tropical and temperate sites—and why these sites are hypothesized to have differently assembled/functioning trophic niches. This is stated in quite general terms—but, it seems a mechanistic basis for the study could be more clearly outlined. Along these lines, the identity of the 7 baits (it is not sufficient to refer to the Houadria paper, Line 137), and the rationale behind them are unclear (e.g. line 132)? Each resource should be clearly linked to an a priori hypothesis, established in the Introduction. For instance, is it really surprising that ‘seeds and melezitose changed positions between communities’ (line 262-263), or that ants used sugars ‘indiscriminately’ in Germany, but not Brazil (lines 263-268)? Was this unexpected?

Reviewer 2 ·

Basic reporting

The authors did a good job on basic reporting and providing their raw data. If it is possible to spell out at least one of the 3 acronyms used in the paper, that would be helpful.

Experimental design

I have three major issues with the experimental design:

1) Storing samples in EtOH changes their carbon signature, and as such shouldn’t be used or discussed. Carbon signatures in Drosophila were lowered with ethanol preservation (Sarakinos et al 2002). I understand that one previous study pointed out (without statistical support) that this not of concern for ants (Bluthgen et al. 2003), but this finding was based on a single temperate zone species and it’s unlikely to hold across all ants. I think this is especially concerning because you show differences in the tropical and temperate zone ants with respect to fatty acids. Why would you then expect that EtOH has the same effect on species from different regions. Without knowing more about this particular effect, I believe C data should be removed from the manuscript. Furthermore, no baseline data was provided, so authors can’t discuss trophic levels as they don’t know with which 15N value does their first tropic level start.

2) Fatty acids were extracted from the whole body, unlike stable isotopes which were extracted from ants after removing the gaster. How can those data be comparable, if you are analyzing chemistry of different body parts? Additionally you collected the ants for fatty acid analysis from the baits. Is it possible that the consumption of the baits changed FAA signature?

3) Overall methods need to be more detailed. No one can replicate this study by reading the methods. The authors are relying on the assumption that the reader is familiar with their recent publications, if not, one needs to download 2 separate publications and read their methodology. I understand you used the same set up, but couldn’t you at least provide some basic information to make the methods more useful.

Validity of the findings

I believe this is a valuable data set, which includes a novel approach, however my concern is that part of the results should not be reported or discussed, and some might not be comparable because of the sampling method (see experimental design for details).

It’s unclear why were particular NLFA chosen to be plotted and discussed in PCA (Figure 4). Specifically Figure 4b does not show the most abundant NLFA: C18:1n9, and it is even present in the tropics in high concentrations. Wouldn’t it be interesting to see if they correlate with the same bait preference in both temperate and tropical regions? I think you should add NLFA recorded in highest concentrations to your PCA, especially if they are recorded in both sites, for example C18:1n9,C16:0, C18:0,.

Lastly, authors conclude that using multiple methods provides a more comprehensive picture of resource use, which is not shown, especially if we look at the temperate data set. It remains unclear how did FAA contribute more to understanding what is going on with resource consumption.

Additional comments

Study by Rosumek et al. explores a topic of resource competition combining an impressive number of resources and techniques using an omnivorous taxon, which are usually underrepresented in studies on resource use. The study tests the difference between the temperate and tropical ant communities in resource use, stable isotope and fatty acid signatures. The authors do a good job of compiling both behavioral and chemical data from two different ant communities, and this kind of data makes this an interesting and novel study.

Minor comments:
L44-6 Correlated in what way? Please be more specific when describing the relationship
L50-51 “Temperate species had distinctive NLFA patterns that were not related to different
diets, but may be influenced by environmental factors” The wording here is problematic because in L44 you talk about NLFA being correlated with bait use. I’m assuming the L50 refers to trophic position of the ants? However, this needs to be clearly stated.
L61-61 This sentence was already used in the abstract – please re-word
L78 Clarify what you mean by environmental mechanisms of coexistence
L81 This is not completely true see Davidson 2003, Bluthgen et al. 2003
L119 Which particular patterns are you referring to? Competition? Resource partitioning?
L134 How big were your grids? What was the total number of baits offered. One bait per point means 1 bait type, or one of each bait types?
L143-145 How many vials were used? How were they placed? How much propylene glycol was used per vial?
L147 Why are some ants used from some baits and not the others? Can fatty acid content be changed with food uptake? If so, how will bait consumption change your NLFA signature. Why aren’t NLFA at both sites sampled from the colonies, and how might that affect your results?
L151 You at least need to mention what solvent was used for your extractions, if nothing else
L155 Preservation of ants in 70% EtOH can alter carbon content. This needs to be addressed.
L194 What was PAST used for?
L262 If higher d’ value refers to the resource d’, then Feces has the highest d’ value in Brazil, higher than Large prey, and yet groups together with small prey, not separated as large prey in Figure 1. Perhaps providing more detail on which data was used to produce these clusters in the methods would help explain this clustering. In methods you say proportion of occurrence was used for Bray Curtis Analysis, but it’s unclear are you referring to the species level occurrence, or overall abundance of ants at baits
L423 Fluidity of what?
L425 If this fatty acid can be synthesized, why would ants store it throughout the year, and not just synthesize it in the fall?
L430 Here would be useful to see if this same NLFA was related to dead arthropod prey in the temperate zone as well. If not, then you can’t be using this paragraph L430-439 as an explanation.
L571 This is becoming repetitive, you already discussed distinct position of Wasmannia.
L594 How do you interpret high 15N then?

Overall I think the discussion would benefit from shortening, maybe even restructuring.

---

## Round 0.2 · accepted · Accept

Normally I'd send the manuscript back to the reviewers after a first decision as "major revisions". However, after reading the rebuttal letter, I believe that you have properly addressed all of the concerns raised by the reviewers.

#